# Landscape-explicit phylogeography illuminates the ecographic radiation of early archosauromorph reptiles

Joseph T. Flannery-Sutherland [1,2] ✉, Armin Elsler [2],
Alexander Farnsworth [3,4], Daniel J. Lunt [3] & Michael J. Benton [2]

Spatial incompleteness in the fossil record severely diminishes the observed ecological and geographic ranges of clades. The biological processes shaping species distributions and richness through time, however, also operate across geographic space and so clade biogeographic histories can indicate where their lineages must have successfully dispersed through these sampling gaps. Consequently, these histories are powerful, yet untapped tools for quantifying their unobserved ecographic diversity. Here, we couple phylogeographic modelling with a landscape connectivity approach to reconstruct the origins and dispersal of Permian–Triassic archosauromorph reptiles. We recover substantial ecographic diversity from the gaps in their fossil record, illuminating the cryptic first 20 million years of their evolutionary history, a peak in climatic disparity in the earliest Triassic period, and dispersals through the Pangaean tropical dead zone which contradict its perception as a hard barrier to vertebrate movement. This remarkable tolerance of climatic adversity was probably integral to their later evolutionary success.

Changing climate across space and through time is a fundamental control on the dispersal, extinction and divergence processes that shape species distributions and diversity patterns[1–3]. Fossil record incompleteness, however, severely challenges climate-driven palaeobiological narratives by simultaneously distorting their observed stratigraphic durations, geographic ranges and environmental tolerances in deep time[1,4,5]. In the face of these spatiotemporal gaps, palaeontologists have explored extrapolative statistical techniques, such as species distribution modelling and ancestral state estimation, to reconstruct deep-time geographic ranges and climatic tolerances. Their use is limited, however, if fossil data are insufficient to adequately characterize a clade's climate space occupancy (climatic niche)[6–12]. Climate space occupancy, however, is inextricably linked to the biogeographic distribution (biotope) of a clade through time[13,14], including within

spatial gaps in the fossil record where they may never be sampled, but through which they must have successfully dispersed. Here, we posit that this ecographic reciprocity between niche and biotope, known as Hutchinson's duality[15,16], makes the phylogeographic history of a clade an informative, yet underappreciated, tool for inferring ancestral climatic tolerances on the basis of how those spatial relationships are embedded within, and connected through, climate space.

We explore the use of Hutchinson's duality for reconstructing clade ecographic diversity using the phylogeographic history of early archosauromorph reptiles. Archosauromorphs (including crocodiles and birds today, as well as the dinosaurian ancestors of the latter) have shown immense taxonomic and ecomorphological diversity across the terrestrial, freshwater and marine realms from the Triassic period to the present[17–21]. Most of their diversity is contained within the bird-line

[1]School of Geography, Earth and Environmental Science, University of Birmingham, Birmingham, UK. [2]School of Earth Sciences, University of Bristol, Bristol, UK. [3]School of Geographical Sciences and Cabot Institute, University of Bristol, Bristol, UK. [4]State Key Laboratory of Tibetan Plateau Earth System, Environment and Resources (TPESER), Institute of Tibetan Plateau Research, Chinese Academy of Sciences, Beijing, China. ✉e-mail: j.t.flannerysutherland@bham.ac.uk

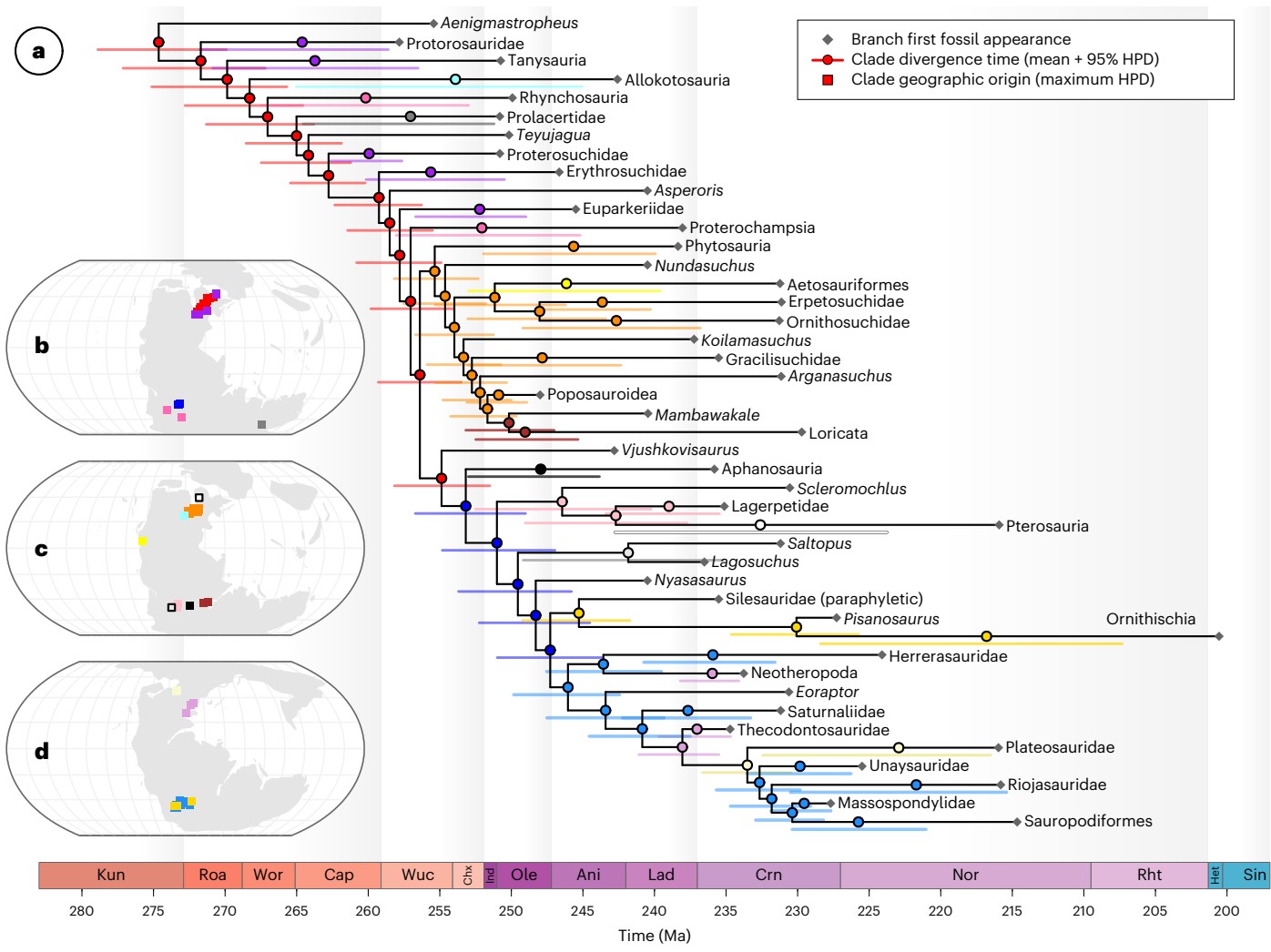

**Fig. 1 | Spatiotemporal origins of archosauromorph higher taxonomic diversity. a–d**, A higher taxonomic summary of a time-calibrated phylogeny of 392 early archosauromorph reptiles, showing their divergence-time uncertainties (**a**) and inferred ancestral origination points (**b–d**). For branches representing higher clades, the tip represents the oldest known fossil of that clade, while the clade origin itself is marked by the coloured circle. Refer to Fig. 3 for the complete tree. Maps display the stages Wuchiapingian (**b**), Induan (**c**) and Norian (**d**), respectively. Ani, Anisian; Cap, Capitanian; Chx, Changhsingian; Crn, Carnian; Het, Hettangian; Ind, Induan; Kun, Kungurian; Lad, Ladinian; Nor, Norian; Ole, Olenekian; Rht, Rhaetian; Roa, Roadian; Sin, Sinemurian; Wor, Wordian; and Wuc, Wuchiapingian. HPD, highest posterior density; Ma, million years ago.

and crocodile-line crown groups (Archosauria) but was founded on the initial adaptive radiation of early-diverging archosauromorphs. Following their cryptic origin in the Middle Permian, archosauromorphs entered a host of ecological niches left vacant by the end-Permian mass extinction[22] before explosive diversification of the crown groups during the Carnian Pluvial Episode in the Late Triassic[23]. In turn, the success of avemetatarsalians (bird-line) compared to pseudosuchians (crocodile-line) has been attributed to their capacity to tolerate a much wider range of climatic conditions[10,24]. These scenarios, however, are based on inferences from their sporadically sampled fossil record and early-diverging phylogenetic relationships. Further, although early archosauromorph phylogeny is broadly well-resolved[25–27], recent fossil discoveries and re-analysis of enigmatic taxa with modern imaging techniques have spurred revisions to placements of many early avemetatarsalian clades[27–36]. Consequently, archosauromorph biogeographic origins and ecographic patterns remain incompletely understood, making them an ideal case study for this work.

To reconstruct early archosauromorph ecographic diversity, we couple Bayesian phylogeographic inference of their origins with an approach for inferring spatially explicit dispersal routes based on landscape connectivity[37–39] termed TARDIS (terrains and routes directed in space–time). TARDIS represents palaeogeographic surfaces and their substantial changes in topography and continental configuration as a flexibly weighted spatiotemporal graph. We estimate the dispersal routes between ancestor and descendant locations in space and time within this graph as least-cost paths whose geometries provide conservative, yet highly informative estimates of the lineage geographic distributions necessitated by their phylogeographic history. These paths then permit measurement of the environmental conditions that clades must have tolerated during dispersal, including through spatial gaps in their fossil record. As spatially resolved climate data based on lithological proxies are also afflicted by geographic incompleteness in the wider geological record, we instead leverage climate reconstructions from recent advances in deep-time earth system modelling[2,40,41]. By connecting the fragmented portions of the early archosauromorph fossil record through unsampled regions of geographic space, we transform inaccessible portions of their biogeographic history into rich sources of data on the breadth of their occupied climate space (climatic disparity), enabling us to estimate the tempo and mode of their early climatic evolution in unprecedented detail.

## Results

### Spatiotemporal origins

We time-calibrated a phylogeny of early archosauromorphs[42,43], then used this tree to estimate their point-wise ancestral geographic origins using the geo model in BayesTraits (Fig. 1 and Supplementary Information). The backbone of higher archosauromorph diversity originated cryptically through the Kungurian to Wuchiapingian stages of the Permian within the European region of northern Pangaea (Fig. 1a,b). Early-diverging archosauromorph and archosauriform clades whose younger fossil records show substantially broader geographic distributions spanning northern Pangaea and southern Gondwana generally conformed to this origination pattern (erythrosuchids, euparkeriids, proterosaurids, proterosuchids and tanysaurids). Some clades dispersed more widely, however, despite their sister and ancestral nodes originating in northern Pangaea (prolacertids in eastern Gondwana, proterochampsids and rhynchosaurs in western Gondwana). Originations within the archosauromorph crown group are sensitive to placement of clades historically viewed as stem dinosaurs (Fig. 2). Revisions of lagerpetids as the sister group of pterosaurs following recent consensus[27,28,31,33] and of silesaurids as an early-diverging grade of ornithischian dinosaurs[29,30,32,34–36] collapse poorly resolved bimodal estimates for the geographic origins of archosaurs and various pseudosuchian clades to unimodal estimates (Fig. 2a,b and Supplementary Information) and substantially reduce divergence-time uncertainties for pterosauromorphs, pterosaurs and ornithischians (Fig. 2c,d). All subsequent results presented here are based on a topology with these revisions.

Under our favoured topology, we recovered a Late Permian origin for pseudosuchians in the northern Pangaean cradle of the wider early archosauromorph radiation (Figs. 1a and 2b). Most early-diverging pseudosuchian clades conformed to this ancestral pattern but there was then a notable southward shift in the locus of crownward pseudosuchian divergences with the origins of loricatans in the early Olenekian stage, while aetosaurs originated around the western seaboard of North America in the early Anisian stage (Fig. 1a,c). Avemetatarsalians also originated in the Late Permian in Europe, preceding a shift to South America with the origin of ornithodirans in the Changhsingian stage (Fig. 1a,b). Higher-level avemetatarsalian divergences proceeded in rapid succession from the Early to Middle Triassic within this new ancestral hub (aphanosaurs, dinosaurs and lagerpetids), while pterosaurs originated >10,000 km away in northern Pangaea during the Carnian stage (Fig. 1a,c). Higher dinosaurian divergences (ornithischians, theropods and sauropodomorphs) also took place in western Gondwana with successive early sauropodomorph clades emerging here through the late Ladinian to late Norian stages (Fig. 1a,d). Conversely, divergences between theropod subclades shifted in the early Carnian from western Gondwana, where the early-diverging herrerasaurids originated, to tropical northern Pangaea with the appearance of coelophysoids and neotheropods (Fig. 1a,d).

### Landscape-explicit dispersal routes

We used landscape connectivity analysis to conservatively estimate the set of dispersal pathways required to connect our point-wise ancestral origin estimates across geographic space and through geological time. Landscape connections were weighted to penalize travel through regions with climatic conditions deviating from those at the ancestral and descendant geographic locations (Methods) while still permitting routes to encompass any expansions of niche space required by their phylogeographic history, modelling the effects of niche conservatism during dispersal.

Phylogeographic rates of dispersal are bimodally distributed with most in the range 100–1,000 km Myr⁻¹, tailing off steadily towards higher orders of magnitude and separated from a second peak close to zero by a trough of dispersal rates ranging from ~5 to 50 km Myr⁻¹ (Fig. 3). There is little apparent clade-wise structuring of rates, although

some groups and key nodes do display extremely low dispersal rates, including throughout heterodontosaurids, early-diverging pterosaurs, proterochampsids, some lineages of aetosaurs and around the origins of archosauriforms, paracrocodylomorphs and sauropodomorphs (Fig. 3). Early archosauromorph dispersal pathways (Fig. 4 and Supplementary Information; animation in Supplementary Video) show a tight spatial distribution amongst oldest divergences and dispersals (Kungurian to Roadian stages) within their northern Pangaean cradle (Fig. 4a). Long-distance dispersals then began from the Wordian stage onwards (Fig. 4a,b,d). These pathways generally took conservative routes along the western Tethys and Zechstein coastlines, but some archosauromorphs notably began crossing the northern branch of the Central Pangaean Mountains into North America in the Capitanian and across the continental interior into South America in the Wordian. Transcontinental traversals continued through the Induan and Olenekian stages of the Early Triassic (Fig. 4b), establishing a second hub of diversification in Gondwana south of the arid belt spanning the austral subtropics, alongside the first transoceanic traversals into East Asia (Fig. 4b). During the Anisian to Carnian, exchanges continued between the northern and southern hemispheres through the corridor situated between the Central Pangaean Mountains and the western Tethys coast (Fig. 4c,d), but also between European and North American Pangaea through the continental interior with increasing frequency through the remainder of the Triassic (Fig. 4d,e).

### Archosauromorph climatic disparity

We constructed a climate space of early archosauromorphs using principal components analysis (PCA) (Fig. 5), based on the climatic conditions simulated at fossil-tip locations from general circulation models. We then estimated ancestral climatic tolerances through their phylogeny using a standard maximum likelihood analysis and projected these estimates into our climate space (ancestral state reconstruction in Fig. 5). Finally, we measured the climatic conditions at our inferred ancestral origination points (ancestral location in Fig. 5) and along their dispersal pathways (ancestral path in Fig. 5), then projected these measurements into climate space to ascertain how much ecographic diversity is recovered when the reciprocity between niche and biotope under Hutchinson's duality is applied through their reconstructed phylogeographic history.

The projected dispersal pathways show wider extent and denser occupancy of climate space by archosauromorphs (Fig. 5e–p) compared to their observed fossil distributions and ancestral state estimates (Fig. 5a–d). During their constrained phase in the Kungurian to Roadian stages, explorations of climate space were not extensive (Fig. 5e), until increased dispersal in the Wordian and Capitanian stages resulted in traverses through environments with high temperature seasonality but low annual precipitation (Fig. 5f), corresponding to the arid subtropics. Through the Late Permian to Early Triassic, archosauromorphs then traversed through climatic conditions nearly as varied as their total climate space occupancy from their origin to the end of the Triassic (Fig. 5g–i). Wide-ranging climatic tolerances persisted through the Middle Triassic and Carnian (Fig. 5j–m), while the Norian and Rhaetian stages were marked by dispersals through environments with high mean annual temperatures and low temperature variation across a range of precipitation seasonalities (Fig. 5n–p).

To quantify the diversity of archosauromorph climatic niches through time, along with how consistently they occupied these niches, we measured the extent of occupied climate space (sums of ranges and variances), the density of occupation within that extent (mean pairwise distance) and the central tendency of occupation (displacement from the centroid). Patterns of climatic disparity recovered from their full phylogeographic history (dispersal paths under Hutchinson's duality) are greatest, followed by disparity inferred from their biogeographic origins (tip and ancestral locations under Hutchinson's duality), while maximum likelihood estimates based on tip states alone (no duality) are

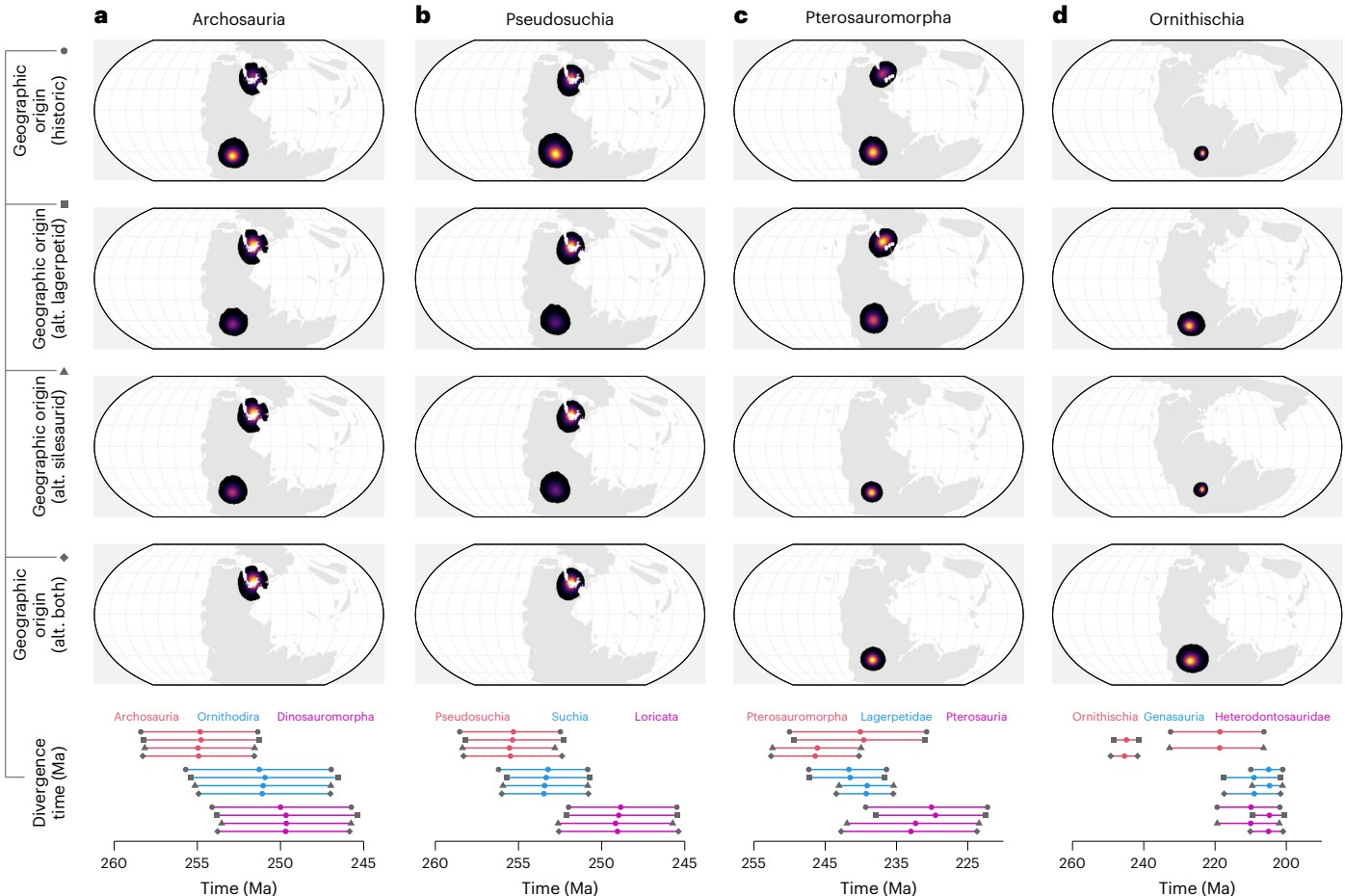

**Fig. 2 | Impact of lagerpetids and silesaurid phylogenetic affinities on archosaurian spatiotemporal origins. a–d**, Heat maps of the 95% highest posterior densities for the geographic origins of archosaurs (**a**), pseudosuchians (**b**), pterosauromorphs (**c**) and ornithischians (**d**); and the 95% highest posterior densities for their divergence times (Ma), as well as the divergence times of key associated nodes, under four alternative (alt.) affinities for lagerpetids and silesaurids: lagerpetids and silesaurids as monophyletic stem dinosaur clades; lagerpetids as the sister group to pterosaurs; silesaurids as a paraphyletic grade of early-diverging ornithischians; and lagerpetids and silesaurids both in their alternative positions. The symbols beside the row labels (historic, alt. lagerpetid, alt. silesaurid and alt. both) correspond, in the same order, to the lines representing the divergence times for each phylogenetic scenario along the bottom row of the plot.

consistently the most conservative (Fig. 6). The different approaches recover trends that are broadly similar through the entire study interval but differ markedly in their finer detail, alongside additional variation between summary metrics. There is a consistent pattern of steadily increasing climatic disparity through the Middle and Late Permian. Maximum likelihood estimates continued rising to an initial peak around the Early/Middle Triassic boundary, then declined through the Middle Triassic. Conversely, dispersal path estimates peaked around the end-Permian mass extinction, with these peaks representing the time of maximal climatic disparity in the study interval, then fluctuated through the Early and Middle Triassic without an overall declining trend. Sum of ranges under all measurement methods declined in the early Carnian (Fig. 6b), while decline occurred for sum of variances (Fig. 6a) and mean pairwise distance (Fig. 6c) from ancestral locations and dispersal pathways, but remained steady for maximum likelihood estimates. Extent and dispersion metrics from ancestral locations and dispersal pathways declined to minima in the late Norian, while maximum likelihood estimates instead fluctuated through this interval without an overall declining trend. Finally, the extent and dispersion metrics from all estimation methods rose through the Rhaetian stage, although the sum of ranges from dispersal pathways increased in a sawtooth manner, rather than smoothly. Conversely, central tendency (displacement from centroid) showed an inverse trend to extent and

dispersion, with displacement steadily declining through the Middle Permian then remaining consistently low through to the Early Jurassic (Fig. 6d). Compared to the previous metrics, all three methods of measuring disparity return a similar degree of displacement, although estimates from dispersal pathways and ancestral locations are slightly greater in the late Norian to Rhaetian (Fig. 6c).

## Discussion

A European origin for archosauromorphs is congruent with the location of their oldest fossils, predating these appearances by ~14 Myr in line with previous analyses[27]. The lag between the origin of a clade and its oldest fossils is an expected feature of incomplete sampling[44], an effect exacerbated when a clade is geographically restricted and so limited in population size. Clustering of dispersal pathways during this unsampled interval potentially indicates that archosauromorphs were constrained for millions of years before their first wide-ranging dispersals across Pangaea in the Wordian (Fig. 4a). Our results could subsequently be interpreted at face value as capturing the genuine signal of their earliest biogeographic history. Conversely, previous work has inferred increasing amniote dispersal throughout the Kungurian and Middle Permian[45] in addition to description of archosauromorph fossil material from the Middle Permian of Brazil[46], tens of thousands of kilometres away from our inferred European origin, although its

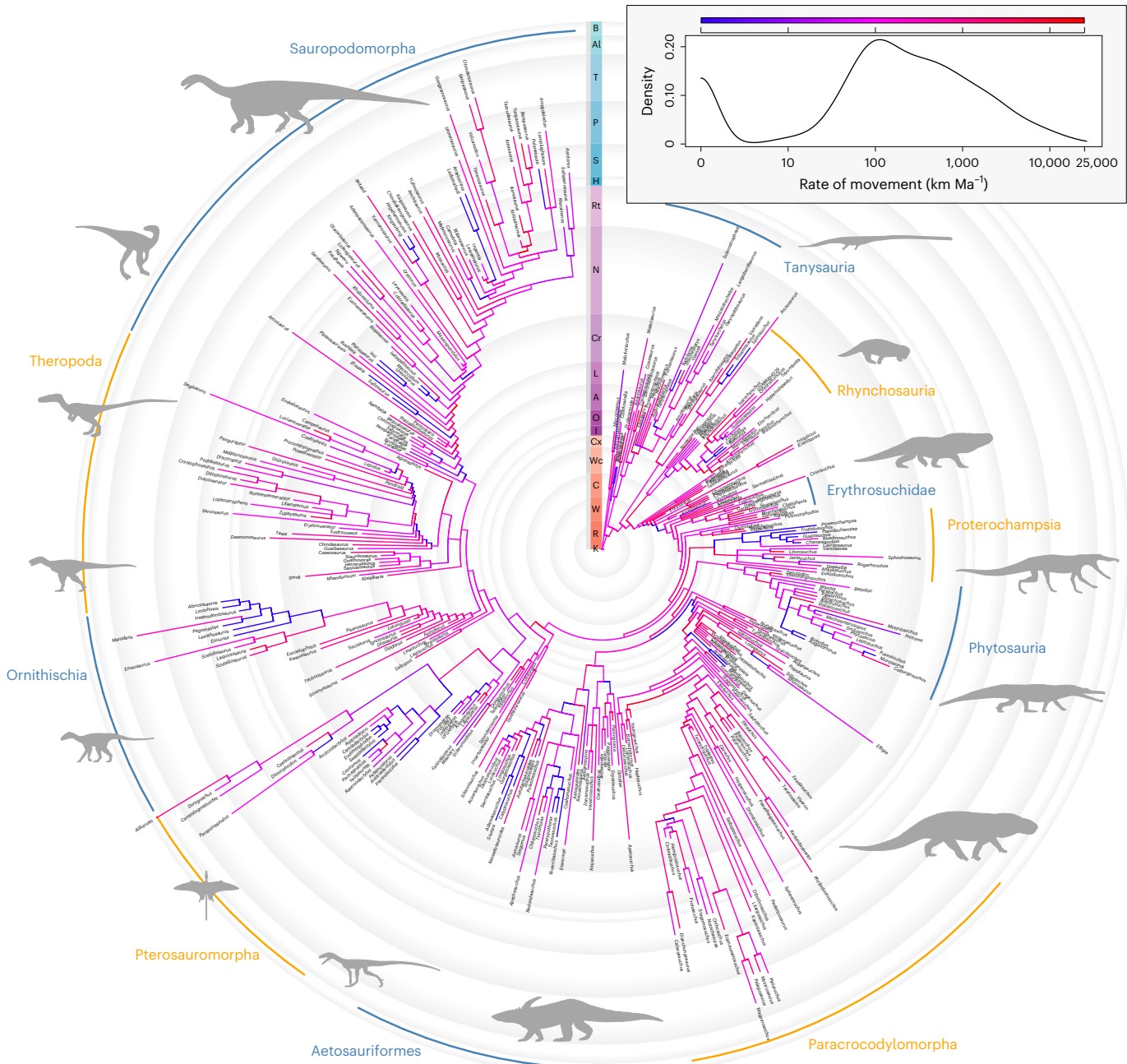

**Fig. 3 | Dispersal rates of early archosauromorphs based on landscape connectivity.** A phylogeny of 392 early archosauromorph taxa coloured according to the geographic dispersal rates along each branch. Lagerpetids are placed as the sister group to pterosaurs and silesaurids as a paraphyletic grade of early-diverging ornithischians. Rates were calculated as geographic pathway lengths between ancestor–descendant origination points modelled using a landscape connectivity approach, divided by the temporal durations of each corresponding phylogenetic branch on the basis of median node divergence times. Major clades of early archosauromorphs are highlighted. Abbreviations for stages: A, Anisian; Al, Aalenian; B, Bathonian; C, Capitanian; Cr, Carnian; Cx, Changhsingian; H, Hettangian; I, Induan; K, Kungurian; L, Ladinian; N, Norian; O, Olenekian; P, Pliensbachian; R, Roadian; Rt, Rhaetian; S, Sinemurian; T, Toarcian; W, Wordian; and Wc, Wuchiapingian. Silhouettes from Phylopic under a Creative Commons license CC0 1.0: *Tanystropheus longobardicus*, *Desmatosuchus* and *Peteinosaurus zambellii*; CC BY 3.0: *Garjainia madiba*, Mark Witton; *Prestosuchus chiniquensis*, Demitry Bogdan; *Melanorosaurus readi*, *Paleorhinus* and *Ixalerpeton polesinensis*, Scott Hartman; *Silesaurus opolensis*, Mathew Wedel; CC BY-SA 3.0: *Chanaresuchus bonapartei*, Smokeybjb; *Coelophysis bauri*, Emily Willoughby; *Eoraptor lunensis*, Marmela; CC BY 4.0: *Hyperodapedon huxleyi*, Nobu Tamura; *Herrerasaurus Ischigualasto's*, Ivan Iofrida.

indeterminate nature precluded inclusion in our trees. Further, analysis of the global preservation potential of tetrapods in contemporary settings shows that the likelihood of fossilization of terrestrial taxa is heavily dependent on their spatial overlap with areas of active sedimentation[47]. As such, the geographic origin and apparent constraint of the earliest archosauromorphs we recover could be artefacts of simply where there was sufficient net sedimentation for archosauromorphs to

first enter the fossil record, overprinted by the distribution of exposed fossiliferous Permian strata in the present.

Elsewhere in our tree, subclade ancestral origin estimates are concordant with the findings of previous authors using a variety of alternative methods, such as a European origin for phytosaurs[48] and a South American origin for dinosaurs[49,50]. There is also a conspicuous correspondence, however, between major areas of fossil sampling and

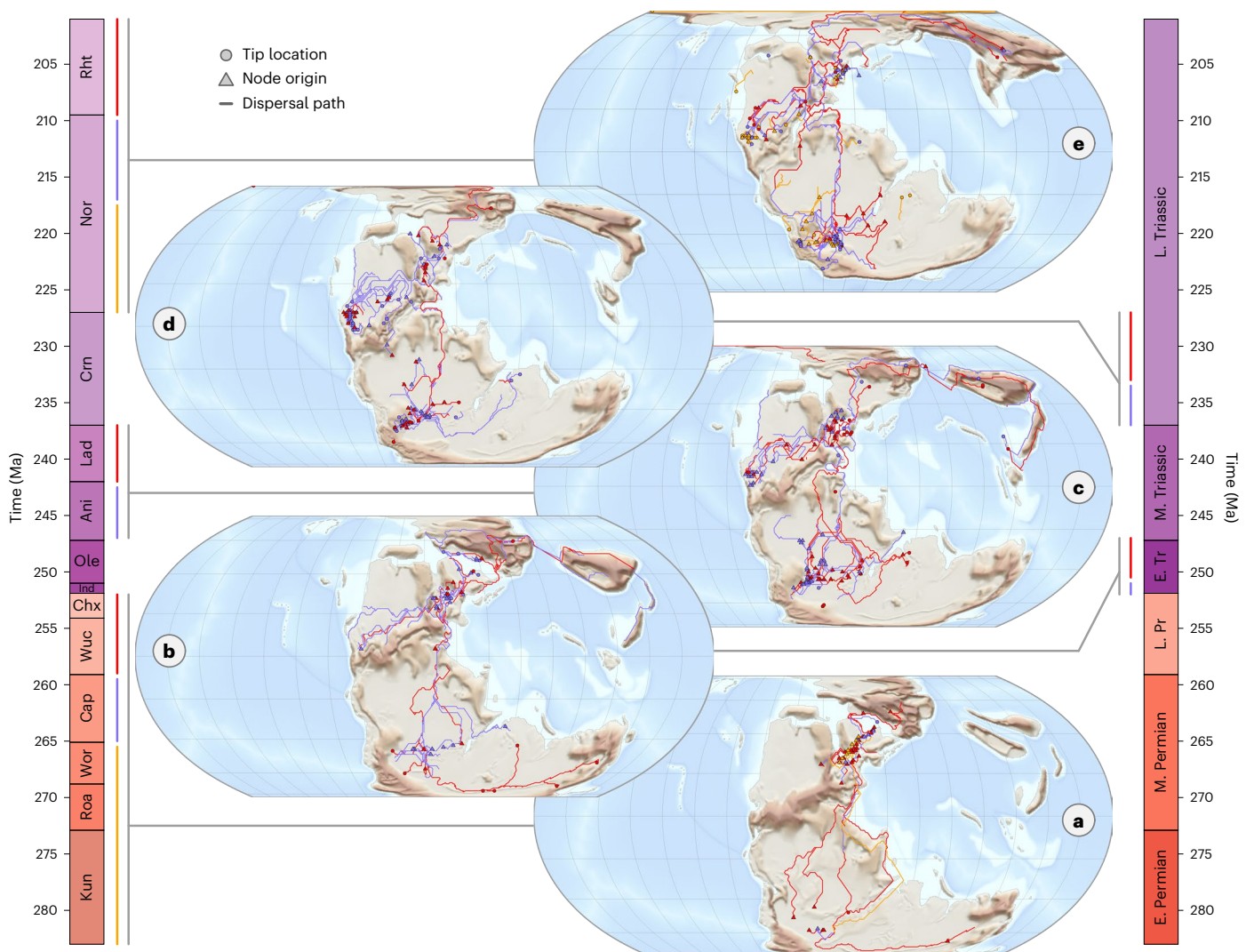

**Fig. 4 | Dispersal routes of early archosauromorphs based on landscape connectivity.** Time slices from a phylogeny of 392 early archosauromorph taxa, as shown in Fig. 3, projected into geographic space as sets of ancestor–descendant dispersal routes modelled using a landscape connectivity approach. In each map, branches are coloured by stratigraphic interval. Open-ended branches are a result of time slicing and should be interpreted as connecting to open ends in other bins, respectively the Permian (**a**), Early Triassic (**b**), Middle Triassic (**c**), Carnian (**d**) and the remainder of the Triassic (**e**). Supplementary figures are available for each individual path which show path directionality (Supplementary Information), in addition to an animation of the pathways (Supplementary Video). E. Permian, Early Permian; M. Permian, Middle Permian; L. Pr, Late Permian; E. Tr, Early Triassic; M. Triassic, Middle Triassic; and L. Triassic, Late Triassic.

inferred divergence locations throughout early archosauromorph phylogeny (Fig. 4a–e). The European fossil record is disproportionately well sampled compared to the rest of the globe[51], with rich stratigraphic successions that cover the Permian and Triassic. From the perspective of sampling bias, it is perhaps unsurprising that many of the oldest known archosauromorph fossils have been found in Europe, inducing geographic origins and deepest phylogenetic splits in northern Pangaea given the rapidity of early divergences along the path towards crown archosaurs. In turn, once geographically closer sedimentary sequences spanning Argentina (Uspallata Supergroup), Brazil (Santa Maria Supergroup) and southern Africa (Karoo Supergroup, Manda Beds) and the extensively sampled Chinle Formation and Dockum Group in the southern United States may be responsible for inducing the southern Gondwanan and North American centres of origination in the Early and Middle Triassic, respectively. The geo model itself is also only able to accept a single phylogenetic topology for analysis, so our results do not account for how divergence-time uncertainties for many of the deepest branching nodes in our tree (~8–10 Myr)

may affect inferred geographic origins. Consequently, inferred clade divergence locations may largely follow spatial patterns dictated by the geographic distributions of their oldest fossils, rather than their true biogeographic origins, making it challenging to assert that the origin of archosauromorphs is genuinely European. Recent work has also found both empirical fossil and model-based support for dinosaurian origins at lower austral palaeolatitudes, including with silesaurids as early-diverging ornithischians[52,53], demonstrating that biogeographic origins clearly display sensitivity to choices of tree topology and ancestral state estimation methods, as well as to the geographic signals imposed by differential taxon sampling between trees.

The oldest fossils of a clade are nonetheless invaluable as the only empirical sources of data on their earliest evolutionary history and it would be premature to treat any investigation of geographic origins in the fossil record as futile a priori, despite the potential impacts of spatial sampling biases. The point-wise estimates from the geo model are also preferable to the potentially vast discrete area estimates returned by other inference methods and are a necessity for

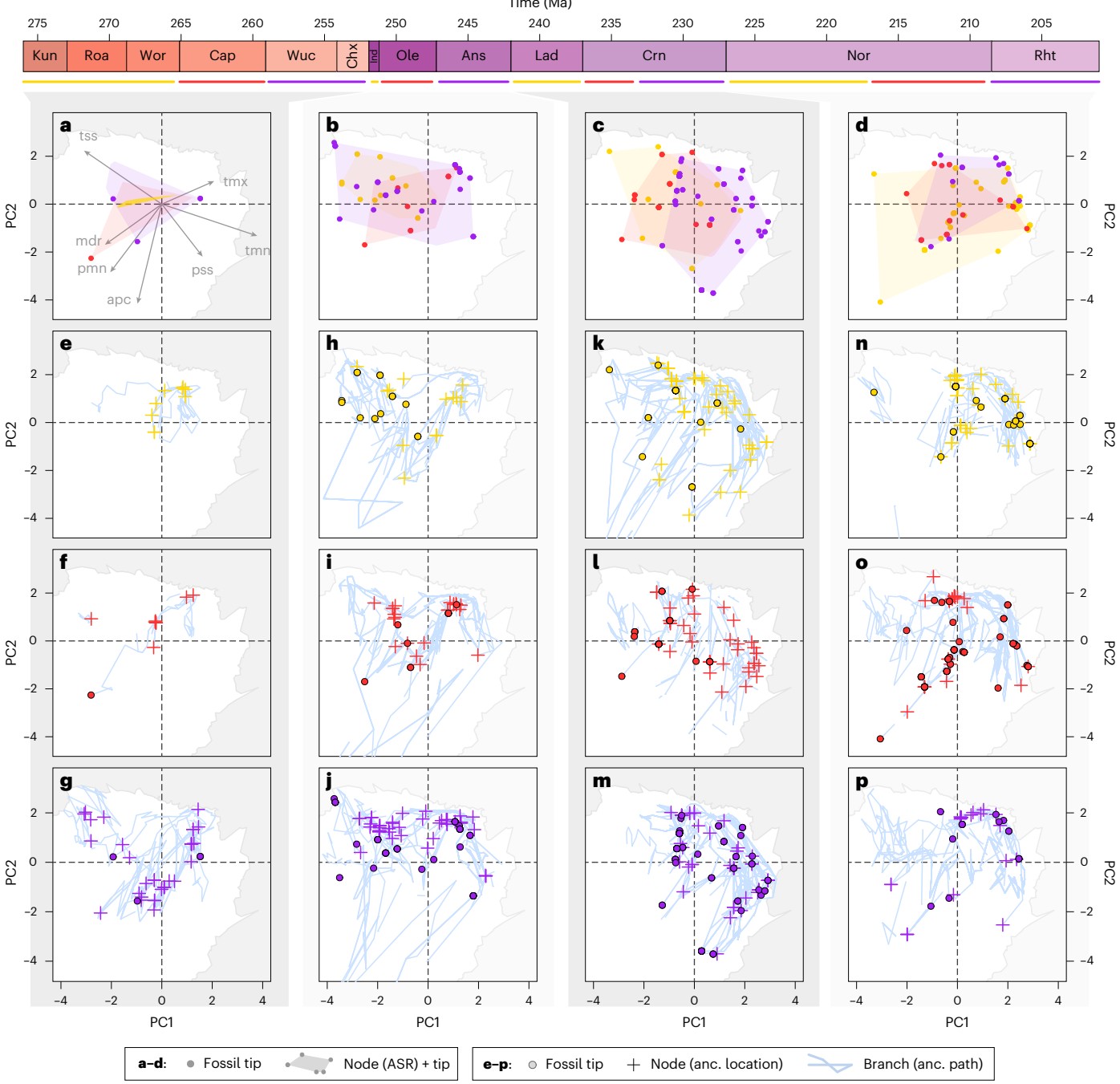

**Fig. 5 | Climate space occupancy of early archosauromorphs. a–p**, PCA of selected climatic conditions (abbreviations in **a**) tolerated by 392 early archosauromorph taxa. Ancestral climate space occupancy was then calculated at stage and substage level by maximum likelihood (**a–d**) or measurement of climatic conditions along ancestral dispersal pathways (**e–p**) shown in Fig. 4, followed by projection into climate space. In each panel, the white region denotes climate space realized on Earth through the stages of the Permian to Jurassic. Open-ended branches are a result of time slicing and should be interpreted as connecting to open ends in other bins. In **a–d**, convex hulls comprise the space occupied by tips and nodes. Nodes have been omitted for clarity as most hulls are composed of tips. In **a** and **b**, however, hull corners without points are where nodes form part of the hull perimeter. Maximum likelihood climate spaces for the Permian (**a**), Early Triassic to Anisian (**b**), Ladinian to Carnian (**c**), and the remainder of the Triassic (**d**), then dispersal path climate space in the Kungurian to Wordian (**e**), Capitanian (**f**), Late Permian (**g**), Induan (**h**), Olenekian (**i**), Anisian (**j**), Ladinian (**k**), early Carnian (**l**), late Carnian (**m**), early Norian (**n**), late Norian (**o**) and Rhaetian (**p**). apc, annual precipitation; ASR, ancestral (anc.) state reconstruction; mdr, mean diurnal range; pmn, precipitation of the minimum quarter; pss, precipitation seasonality; tmn, temperature of the minimum quarter; tmx, temperature of the maximum quarter; and tss, temperature seasonality.

subsequent inference of dispersal pathways and ecographic diversity using landscape connectivity. When niche and biotope are linked together by Hutchinson's duality along these pathways, greater ecographic diversity is recovered for early archosauromorphs compared to what their fossil record or maximum likelihood ancestral state estimates would show (Figs. 5 and 6). These effects are most prominent through the 24 Myr between inferred phylogenetic origin in the latest Kungurian to a widely sampled archosauromorph record from the Early

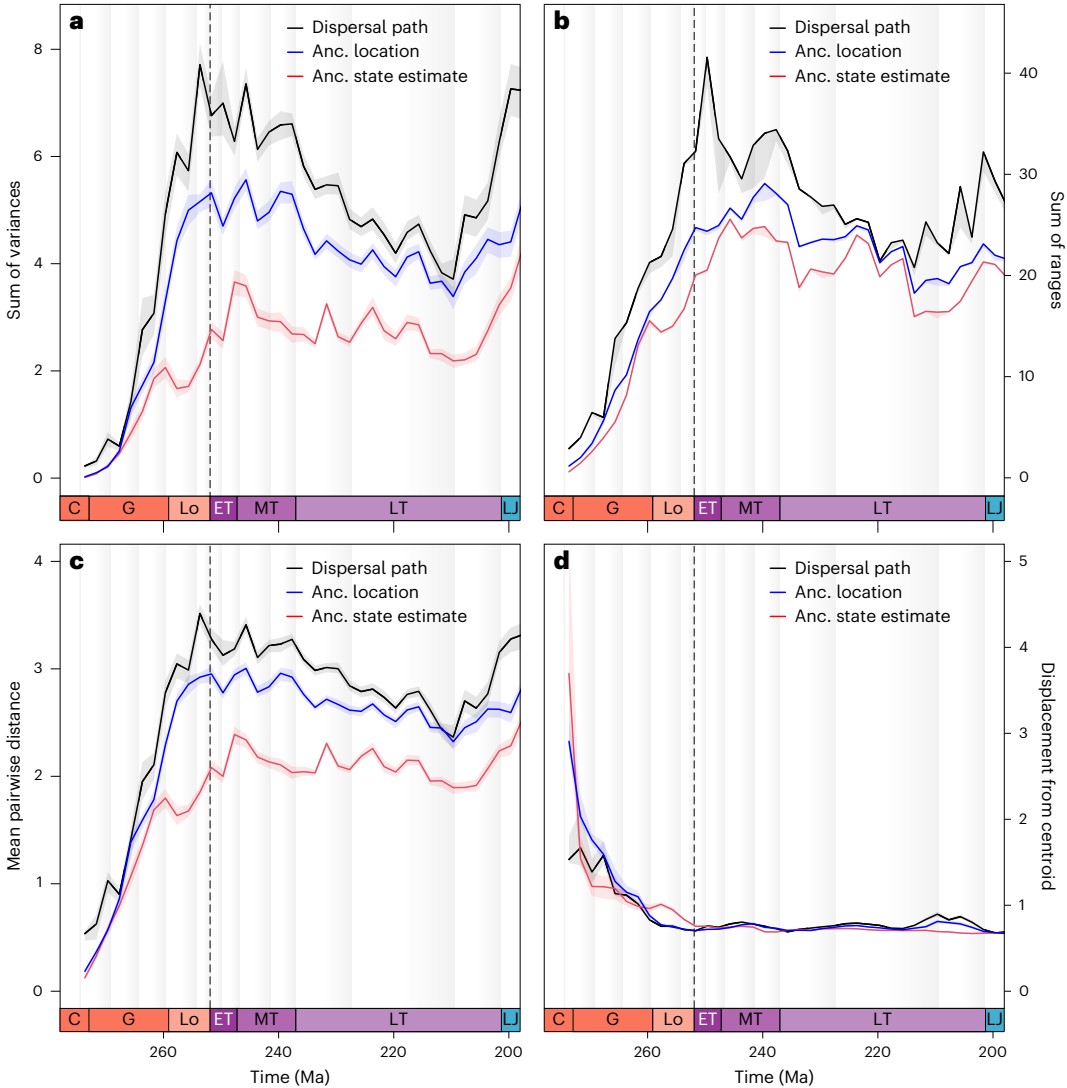

**Fig. 6 | Summary metrics of the diversity and consistency of early archosauromorph climatic disparity through time. a,b**, Sum of variances (**a**) and sum of ranges (**b**) measure the extent of occupied climate space. **c**, Mean pairwise distance measures the density of climate space occupation. **d**, Displacement from centroid measures the central tendency of climate space occupation. Each metric was calculated in 2-million-year time bins for three different disparity datasets: disparity based on maximum likelihood and tip states; disparity based on ancestral geographic locations and tip states; and disparity measured along ancestral dispersal pathways, as displayed in Fig. 5. As such, the second (blue) and third datasets (black) incorporate additional climatic disparity under Hutchinson's duality, compared to the first dataset (red). Each line displays the median disparity with a 95% confidence interval calculated from 1,000 randomized bootstrapped trials. The vertical dotted lines mark the end-Permian mass extinction at the Permian/Triassic boundary. Grey bars denote stage-level chronostratigraphic divisions. C, Cisuralian; ET, Early Triassic; G, Guadalupian; Lo, Lopingian; LT, Late Triassic; and MT, Middle Triassic.

Triassic (Fig. 5e–h), although the extent of climate space occupied by ancestral geographic locations continues to outstrip the extent of fossil tips until the Anisian (Fig. 5I). Occupancy from tip and node locations is more congruent from this point (Fig. 5j–p), possibly due to their more consistent geographic overlap (Fig. 4). As this overlap may also reflect spatial sampling bias in the archosauromorph fossil record, future fossil discoveries from unsampled parts of geographic space may expand ancestral location ranges and climate space occupancy. Validating this hypothesis, however, is reliant on the wider availability of fossiliferous strata representing appropriate environments and stratigraphic ages, highlighting the benefit of using explicit phylogeographic dispersal pathways to conservatively reconstruct ecographic diversity within spatial sampling gaps. Incorporation of climatic conditions encountered during dispersal alters long- and short-term trends in climate space occupancy through niche–biotope coupling. For example, ancestral state estimations suggest that climate occupancy experienced a

transient increase through the Carnian in association with the Carnian Pluvial Episode, while dispersal pathways instead recover progressive decline through the entire stage (Fig. 6). Over longer timescales, dispersal pathways also recover a long-term decline in climate space occupancy from the earliest Triassic to the end of the Norian compared to ancestral state estimates, although this trend reflects an average across the entirety of our phylogeny from which subclade disparities may conceivably depart.

Notably, our dispersal paths extend well beyond the observed bounds of the archosauromorph fossil record in both geographic and temporal terms. While the Middle Permian Brazilian archosauromorph material is indeterminate and does not inform our phylogenetic ancestral location estimates, our dispersal pathways nonetheless model their entry into southern Gondwana in the Wordian (Fig. 4a). Similarly, Capitanian dispersal pathways into North America across the northern branch of the Central Pangaean mountains predate the oldest

known archosauromorph taxon on the continent, the ctenosauriscid *Arizonasaurus babitti*, by nearly 15 Myr and their oldest footprint records by a similar degree[54]. We also recover dispersal pathways spanning the tropics in the immediate aftermath of the end-Permian mass extinction (Fig. 4b). Cluster analysis of Early Triassic tetrapod assemblages has previously indicated the existence of three broad Pangaean bioregions in the Early Triassic: Northern European, Gondwanan and Euramerican, with the last two supposedly separated by lithologically inferred, thermophysiologically hostile climatic conditions of the 'tropical dead zone'[55–57]. While these regions may have largely operated as distinct biogeographic units, early archosauromorphs continued to undertake transequatorial traverses along the western Tethys coastline as they radiated taxonomically and ecologically into vacant niches in the Early Triassic. This indicates that the tropical dead zone was not an absolute barrier to dispersal during this interval, a supposition matching the wide geographic range of their Early Triassic trace fossil record[54]. The tropical dead zone may have varied in extent with substage temperature fluctuations, as inferred from lithological proxies, allowing dispersal during cooler times.

The coastal dispersal route recovered in the Early Triassic may have been favourable as a result of its modest topography compared to the Central Pangaean Mountains further east and intermittent monsoonal humidity countering continental aridity, evidenced by alternation between red bed and playa lake facies across several sedimentary basins situated along the palaeocoastline[58,59]. The pathways are also subject to the spatial limits of our landscape data, however, and so may not capture the effects of geographic features at smaller scales. For example, large rivers may have created humid corridors for dispersal through otherwise arid regions analogous to the corridor of habitability surrounding the Nile River through the Eastern Desert of Egypt. Information on the precise geometry of palaeoriver networks on global scales could nonetheless be estimated from catchment geometries delineated from landscape topography using flow accumulation algorithms[60] and incorporated during dispersal modelling. Our approach also assumes a process of dispersal between two points and so does not account for other biogeographic processes affecting populations, such as vicariance[61]. The influence of such processes could potentially be investigated, however, by using community detection algorithms to identify increasing or decreasing biogeographic connectivity between spatiotemporally discrete neighbourhoods within our landscape graphs.

While we use deep-time climate simulations to overcome the major hurdle posed by spatially incomplete sampling of lithological climate proxies, this approach introduces other uncertainties into our niche modelling framework. The spatiotemporal grain of the climate simulations captures long-term environmental averages over scales of hundreds of kilometres, meaning that conditions measured at tip and node locations may be unrepresentative of shorter-term, localized microclimates[62] that may have shaped archosauromorph dispersal. The conditions measured at tip locations must have fallen within the absolute climatic tolerances of our chosen taxa, but our niche modelling approach does not then account for the possibility that these may represent the extremes of their tolerated niche rather than their preferred environmental conditions. In addition, many of these taxa are only known from single fossil sites, yielding point-wise estimates of their climatic tolerances rather than the climatic envelope expected under more complete sampling of their true geographic ranges[8]. In turn, the climatic restrictions we apply during dispersal are potentially overly conservative, necessitating longer dispersal pathways to explain phylogeographic distributions through geological time. As our climatic restrictions are ultimately projected into geographic space during dispersal pathway estimation, however, this opens the potential for using standard statistical techniques from species distribution modelling to evaluate their predictive power. Finally, our approach only models the fundamental niche, excluding any biotic factors that may have

also shaped dispersal, a limitation also noted for dispersal analysis of temnospondyl amphibians in the Triassic[63]. Nonetheless, we expect climatic variables to be first-order controls on dispersal, impacting both the thermophysiology of a dispersing organism and broad geographic distributions of biotic variables that also supported its dispersal, such as the presence of preferred food sources or reproductive partners.

In summary, we recover a European origin for archosauromorphs in the latest Kungurian, followed by geographically widespread dispersals beginning as early as Wordian. Origination events within the archosaur crown group are sensitive to phylogenetic placement of clades historically considered to be stem dinosaurs, but resolution of lagerpetids as pterosauromorphs and silesaurids as early-diverging ornithischians substantially decreases spatiotemporal uncertainty in multiple node origins. There is a conspicuous correspondence between the locations of the oldest fossils of a clade and its inferred spatial origin, suggesting that geographic ancestral state estimations may reflect sampling biases as much as biogeographic history. Nonetheless, our landscape connectivity approach improves the completeness of our view of clade ecographic diversity by enabling sampling of their evolutionary history away from their observed fossil record, even if that dispersal history is impacted by spatial sampling biases. The additional spatial and climatic breadth recovered by these pathways hints at what cannot be observed in the current fossil record in terms of geographic and climate space occupancy, arising from the necessary reciprocity between niche and biotope through their phylogeographic history.

Previous workers have considered how ecological niche modelling may be coupled with ancestral state estimation to model potential ancestral geographic ranges through phylogeny[14,64–66]. These approaches also make use of Hutchinson's duality but effectively project from niche to biotope. Our approach inverts this relationship, using the biotope based on ancestral biogeographic relationships to expand the niche. Future advances, however, may come from development of integrated approaches that jointly model the evolution of ancestral niche and biotope to propose geographic origins congruent with how climatic tolerances are expected to evolve through phylogeny. Finally, landscape connectivity and climate-based approaches may improve our understanding of niche dynamics in the fossil record for other groups with patchy fossil records, but graphical representation of spatiotemporal landscapes for investigation of other facets of deep-time biogeography merits further investigation.

## Methods

### Phylogenetic trees and fossil occurrences

We subsampled a phylogeny of archosauromorph reptiles from an early tetrapod informal supertree[42,43], covering all Permian and Triassic species plus a few Jurassic taxa that subtended Triassic-aged nodes. The tree of ref. 26 forms the scaffold for Archosauromorpha with several formal phylogenies informing subclade-specific topologies, alongside modifications herein to update species-level revisions, add taxa named up to May 2022 and discard tips considered to be nomina dubia or of non-archosauromorph affinity. Species tips were collapsed to genus level owing to ongoing controversies over species-level taxonomy in parts of the tree (for example, *Plateosaurus*), although most are already monotypic. All modifications were conducted in R (v.4.41)[67].

We made several revisions to our source tree to reflect contemporary and alternative hypotheses for two early archosaurian clades. First, we moved lagerpetids from the sister position to dinosaurs to the sister position to pterosaurs, in line with current phylogenetic consensus[27,28,31,33]. Second, we considered alternative phylogenetic placement of silesaurids as a paraphyletic grade of stem ornithischians[29,30,32,35,36]. We favoured the resolved topology of lagerpetids in our source tree[42,43] over the clade-wide polytomy in ref. 27. Specimen-based operational taxonomic units in ref. 29 were added to the silesaurid topology in our source tree, one of which additionally features in ref. 27. We also used the source tree topology for

downstream analysis to elucidate the biogeographic impacts of these major revisions on clades historically viewed as early-diverging dinosauromorphs. While the interrelationships of theropod, ornithischian and sauropodomorph dinosaurs remain contentious[68], we retained the topology of ref. 69 (discussion in ref. 34). We elected to retain the interrelationships of eucrocopodans and early-diverging pseudosuchians in our source tree, although alternative positions have been proposed for phytosaurs and doswelliids[25,70,71]. Consequently, we analysed four topologies: a tree with historic placements of lagerpetids and silesaurids as monophyletic dinosauromorph clades; a tree with the current consensus placement of lagerpetids as sister to pterosaurs; a tree with the debated placement of silesaurids as early-diverging ornithischians; and a final tree with both modifications.

We downloaded fossil occurrence data for all genera in tree set from the Paleobiology Database on 1 August 2022, manually added occurrences for absent taxa, then updated any genus-level synonymies. Chronostratigraphic ages were revised using the primary literature, then updated to GTS2020 standard using the chrono_scale() function of the fossilbrush R package[72]. Palaeocoordinates were calculated from the present-day latitudes and longitudes of each occurrence using Getech PLC plate rotation based on their midpoint ages, with a custom R function. All occurrence and tree data are available in ref. 73.

## Time-scaled phylogenies

Each tree was time-scaled under the fossilized birth–death (FBD) model using the clockless tip-dating method of ref. 74 in MrBayes (v.3.3.7)[75,76]. Input nexus files with blank 'dummy' matrices were created with the createMrBayesTipDatingNexus() function of the paleotree R package[77], using default function settings as these reflect best methodological practise[78,79]. Root age priors were set as an exponential offset by 10 Myr from the maximum age of the oldest fossil observation in the tree, based on estimates of a Middle-Late Permian age for the origin of Archosauromorpha[27]. Tip priors were set as uniform distributions bounded by stratigraphic uncertainties of their oldest occurrences, aside from the youngest taxon in the tree (*Allkaruen*, Pterosauria, Toarcian of Argentina) which was held constant as its maximum geological age to permit post hoc conversion of relative branch lengths to absolute calendar time. Markov Chain Monte Carlo (MCMC) was used to estimate FBD model parameters from their posterior distributions. Four replicate analyses were run for each tree for 250 million generations, sampling every 1,000, each comprising four Metropolis-coupled MCMC chains to more effectively explore parameter space. Model parameters were summarized across all chains, with convergence identified by potential scale-reduction factors approaching one following ref. 80. Similar mean estimated divergence times and their highest posterior densities across all four archosauromorph phylogenies indicate analytical robustness, aside from a few nodes associated with the alternative positions of lagerpetids and silesaurids. We also calculated the stratigraphic consistency index (SCI)[81] and gap excess ratio (GER)[82] for our favoured tree ($t_1$) and for a tree inferred using a Bayesian tip-dating approach ($t_2$)[52], based on the set of tips common to both phylogenies, with their similarity (SCI$_{t1}$ = 70%, SCI$_{t2}$ = 72%, GER$_{t1}$ = 79%, GER$_{t2}$ = 80%) lending further support to our informal topology.

## Geographic origin ancestral state estimation

Rather than use ancestral state estimation methods requiring subjective a priori designation of discrete bioregions, we estimated geographic origins as continuous coordinates using the geo model of ref. 50, which considers range evolution through phylogeny as a random walk across the surface of the Earth. Model parameters (internal node and tip points of origin and dispersal rates) are estimated by MCMC from a time-scaled phylogeny and the palaeolongitude–palaeolatitude coordinates of its tip observations, supplied as singleton observations or sets of observations from which estimated tip states are sampled according to their probability. By default, the geo model

considers all regions of the globe as equally accessible, ignoring how past continental configurations might have limited dispersal, although estimated locations may remain plausible subject to the quality of the phylogeny under consideration[50]. An updated version of the model used here, however, uses sets of geographic masks to designate inaccessible regions within discrete time windows (for example, geological stages)[83]. Currently, the model only considers a single topology, precluding inclusion of phylogenetic uncertainty in an analysis, although polytomies are permitted, with several present in our trees.

Geographic origin ancestral states were estimated for each time-scaled archosauromorph phylogeny using the geo model with geographic restrictions in BayesTraits (v.4.0)[83,84]. Geographic masks were generated from Early Permian (Kungurian) to Middle Jurassic (Bajocian) stage-level palaeogeographic digital elevation models (DEMs) from Getech PLC[85]. Land–sea masks were generated from the DEMs at their native resolution (0.5° × 0.5°), upscaled by a factor of two (0.25° × 0.25°) to improve proposal of continuous coordinates in the geo model analysis, then converted to BayesTraits-compatible mask files. Occurrence palaeocoordinates were checked to ensure that they fell within their stage-specific palaeogeographic masks and adjusted to the nearest land cell where necessary. While the geo model can accept a sample of tip state coordinates, a matching set of single tips is needed to initiate the analysis. These were supplied directly in the case of singleton taxa, as great circle midpoints for pairs of tip observations, or the centroid for three or more tip observations. Two models were fitted for each tree with six replicate analyses per model: one where dispersal rates varied uniformly across the tree according to a Brownian motion prior; the other where variable rates (VR) across the tree were proposed using reverse jump MCMC[86]. Each analysis was run for one billion iterations, discarding the first 50% as burn-in, then sampling every 50,000. During each analysis, stepping-stone sampling was used to calculate the marginal likelihood of the fitted model, with 1,000 equally spaced stones through the postburn-in portion of the chain and 100,000 iterations per stone.

Postprocessing of BayesTraits log files was performed in R. Model marginal likelihoods were averaged across replicate analyses with Bayes factors[87] identifying the VR models as better fitting in all cases (logBF = 8). MCMC traces were inspected to visually assess stationarity, then convergence between replicate analyses tested using the gelman.diag() function of the coda R package[88] to calculate parameter-wise potential scale-reduction factors (PSRFs). PSRFs > 1.1 indicated inadequate convergence between chains for some ancestral state estimates, despite the high run length. Nonetheless, ~90% of model parameters had PSRFs approaching one (Supplementary Information), with 95.5% of terminal nodes and 85.4% of internal nodes displaying unimodal posterior distributions of proposed origin points. Consequently, only a few nodes where singular geographic points of origin could not be readily estimated appear responsible for incomplete convergence in the VR models. We therefore take the VR analyses as largely reliable, given their overwhelming Bayes factor support. Replicate log files were concatenated for each VR analysis, sets of ancestral origination points spatially binned using a 1° × 1° grid and smoothed with a Gaussian kernel, then summarized by taking the cell-centre coordinates of highest density cell in the grid, and the 95% highest density intervals for the estimated palaeolongitudes and palaeolatitudes.

## Deep-time climate simulations

To quantify archosauromorph climatic tolerances, we used climate simulations from a fully coupled atmosphere–ocean general circulation model (GCM) HadCM3L-M2.1aE[89]. The ability of a similar model version to reconstruct temperature patterns of the early Eocene[90] and Pliocene[91] shows that the large-scale features of deep-time climates are well simulated, but some regional disparities remain, particularly insufficiently warm high palaeolatitudes under high $CO_2$. Recent model tuning[92], however, has improved these deficiencies, making it suitable

for our global-scale niche modelling approach. The HadCM3L GCM has a spatial resolution of 2.5° × 3.75° with 19 and 20 vertical levels in the atmosphere and ocean respectively. Sea ice is calculated on a zero-layer model on top of the ocean surface. Vegetation is predicted as a fraction for each grid box using a dynamical vegetation model TRIFFID (top-down representation of interactive foliage and flora including dynamics) in equilibrium mode and the MOSES 2.1 land surface scheme[93]. Stage-specific solar luminosity was calculated following ref. 94. Simulation topographic and bathymetric boundary conditions were taken from stage-level Getech digital elevation models reconstructed under highstand sea level[95] and downscaled from 0.5° × 0.5° to the climate model resolution. Atmospheric $CO_2$ values were taken from the proxy estimates of ref. 96 on the basis of stage midpoint ages. Additional simulations were run for the early and late Carnian and the early, middle and late Norian using the Getech stage-level boundary conditions and substage-appropriate $CO_2$ values.

Each simulation was run for 7,500 model years, having been initialized from a pre-industrial state, to allow surface and deep ocean levels to achieve an equilibrium state with close to zero net energy imbalance at the top of the atmosphere. This is crucial as ocean circulation can take many thousands of model years to establish. Climate means are calculated from the last 100 years of each simulation. Compared to ref. 89, the model used for the simulations presented here also contains several updates, mostly designed to improve the representation of polar amplification in past warm climates, using similar methods to those presented in refs. 92,97. The simulations used are detailed in ref. 73 and archived in the PUMA (providing unified model access) database.

### Ancestral climate tolerance estimation

We investigated the climatic tolerances of early archosauromorphs using the BIOCLIM scheme, a series of 19 biologically relevant climatic variables developed for species distribution modelling in terrestrial habitats[98–100]. These variables are calculated from monthly temperature and precipitation maxima and minima and include common metrics such as mean annual temperature and annual precipitation, along with descriptors of their ranges and seasonal variations. Sets of BIOCLIM layers were calculated from each climate simulation using the bioclim() function of the dismo R package[101]. Variable sets were investigated for multicollinearity using Pearson correlation, dendrograms constructed from all pairwise correlations, clusters of highly correlated variables identified at a dendrogram node height of 0.3 (corresponding to 70% positive correlation; Supplementary Information) and redundant variables within each cluster discarded. We favoured selection of monthly over quarterly variables to capture extremes in climatic conditions. We also discarded isothermality due to severe abnormalities in some layers resulting from the sensitivity of the calculation method to the input climate data, despite it falling below the correlation threshold. In the end, we used mean diurnal temperature range, temperature seasonality, maximum temperature in the warmest month, minimum temperature in the coldest month, annual precipitation, precipitation in the driest month and precipitation seasonality.

The selected climatic conditions tolerated by fossil archosauromorphs were extracted from each set of BIOCLIM layers in R using their geo model geographic origin estimates, which include their occurrence observations. Climatic conditions were spatially sampled from the layer sets with respect to tip age, in proportion to the density of palaeolongitude–palaeolatitude estimates for each tip. Sets of conditions were summarized using their mean and 95% confidence intervals, then ordinated using PCA to construct the observed climate space of early archosauromorphs. Next, we inferred ancestral climate tolerances at phylogenetic nodes in two ways. First, climatic conditions at internal nodes were extracted using geo model geographic origin estimates in the same approach as for the tips. Second, maximum likelihood ancestral state estimates and 95% confidence intervals were calculated for the internal nodes with the fastAnc() function of the phytools R

package[102], using the BIOCLIM variables sampled at the locations of the empirical fossil tips. Both sets of node estimates were then projected into the PCA-derived climate space of the fossil tips.

### Spatiotemporal landscape graphs

Dispersal between ancestor–descendant coordinate pairs from the geo model has previously been approximated using great circle (geodesic) paths[50,83,103,104]. This accounts for the necessity of travel around the curvature of the Earth, but not additional distance incurred through changes in elevation (albeit a second-order effect on continental to global scales) and implicitly assumes that all areas of the globe are accessible with isotropic ease of movement. In reality, geodesic paths may intersect major geographic barriers to dispersal. Instead, we model movement using least-cost paths (LCPs), a tool used in landscape connectivity analysis to represent routes of lowest resistance between locations across an anisotropic landscape. LCPs have been previously applied in ecological problems involving the movement of organisms through spatially heterogenous, anisotropic landscapes[105–107], but only rarely in palaeontological and phylogeographic contexts[62,104,108] as they are calculated on fixed landscapes, while palaeobiological dispersals occur across time spans with substantial variation in landscape architecture.

To estimate phylogeographic paths in a landscape connectivity framework, we present TARDIS, an R package for representing spatiotemporal landscapes as time-ordered graphs. TARDIS graphs are constructed from a time-ordered stack of topographic landscape rasters. Each raster grid is converted to a two-dimensional lattice graph where graph vertices represent raster cells, with graph edges connecting orthogonally and/or diagonally adjacent cells (four-degree Rook's case versus eight-degree Queen's case), including across the antimeridian for global rasters. In an idealized, isotropic landscape, all edges are bidirectional and equally weighted; travel is possible in both directions, incurring the same cost regardless of edge or direction. For anisotropic landscapes, edge weights vary according to the cost of traversal between grid cells and may be asymmetric for a given edge; the cost of transit from $a$ to $b$ is not equivalent to $b$ to $a$, including the scenario that travel may be possible in one direction only. Initially, symmetric edge weights are assigned as great circle distances between adjacent grid cell centres, adjusted using Pythagoras' theorem to account for changes in elevation. This basic geographic representation scheme is generally equivalent to those in other R packages (for example, gdistance[109] and topoDistance[110]).

Regions of each raster can be masked to denote cells which are not traversable (for example, oceans for terrestrial connectivity scenarios). These will not receive graph vertices but may induce inaccessible 'islands' of isolated vertices. To ensure that these remain accessible from one other if desired, additional graph edges can optionally be constructed from the geographically closest vertices between islands. The degree of connectivity is flexible. When equal to one, islands are successively linked by a single edge by their geographically nearest vertices, resulting in a minimum spanning tree (MST), the simplest solution for ensuring connectivity of all landscape patches[37]. MST connectivity, however, might be unrealistically conservative for closely positioned islands so $k$-nearest neighbour linkage can be used instead. The maximum number of links between islands is determined by their Voronoi neighbourhood, ensuring the highest possible connectivity without producing edges which intersect other islands, a solution mostly equivalent to the minimum planar graph concept in the landscape connectivity R package grainscape[111].

Once geographical connections in each two-dimensional lattice graph are constructed, the lattices are sequentially linked in time order to create a three-dimensional lattice graph. By default, 'vertical' unidirectional edges link spatially homologous cells forwards in time between 'horizontal' lattice layers. This is appropriate when geographic locations remain constant within the reference frame of the landscape extent.

For global analyses through geological time, however, homologous geographic locations shift position through continental drift and vertical edges can be adjusted accordingly. Vertical edges spanning all time frames are unweighted so that costs of traversal relate entirely to changes within landscapes, while masked cells do not receive any inbound or outbound temporal edges. Not all cells need to have vertical connections defined, nor will this be possible if masked cells are present, but at least one connection is required between layers to ensure continuous passage of time through the graph.

In the final three-dimensional lattice graph, all movement in geographic space occurs along the horizontal edges, with changes in elevation embedded in their graph weights, while all movement through time occurs along the vertical edges. The LCP between any pair of space–time coordinates can then be found using Dijkstra's algorithm[106,112], provided those coordinates map onto vertices in the lattice graph. The cost of this path will be the shortest geographic distance between those points, accounting for all changes in landscape configuration in the intervening time, although time itself has no effect on the shortest path as all temporal connections have zero cost. The horizontal graph weights, however, can be altered flexibly to impose constraints on movement and dispersal besides geographic distance, including separate weighting for edges connecting islands of vertices. While capturing complex landscape connectivity is desirable, weighting strategy strongly affects LCP calculation and may ultimately be subjective, necessitating careful choice informed by biological realism[113], although the original geographic weights are stored within the graph so that the geographic distance along LCP can be determined. Our weighting scheme is broadly inspired by the methodology of the gen3sis R package[114] with extensions and modifications to support our spatiotemporal landscape representation. All code for the construction and analysis of spatiotemporal graphs is available in the TARDIS R package on GitHub. A static version of the package is also present in ref. 73 to ensure complete reproducibility with the other code and scripts therein.

### Phylogeographic dispersal paths

We constructed spatiotemporal landscape graphs from the set of Getech DEMs used for our climate simulations. DEMs were downscaled from their native resolution to 1° × 1° palaeolatitude by palaeolongitude to circumvent memory limitations and ocean cells masked. Different degrees of island linkage were trialled, with MST linkage (one-degree) selected to minimize cases where LCPs hopped erratically between islands across large regions of ocean. Queen's case (eight-degree) horizontal edges were used within non-masked regions of each DEM. Vertical links through time were calculated between spatially homologous points using the plate rotation model of Getech PLC.

Climate and topography are major ecological controls on dispersal[115]. To account for these effects, the spatiotemporal graph was reweighted for pair of ancestor–descendent nodes. We took the node pair coordinates in our PCA-derived climate space and calculated their intervening lines of multiple regression using all principal component axes. These lines model the shift in climatic niche position between ancestor and descendant as smooth change with minimal variation in principal component values along each branch, equivalent to the gradual model of ref. 116. We then calculated the climatic conditions along each horizontal link in the three-dimensional lattice graph as the means of the BIOCLIM scheme between source and destination cells, transformed those means into climate space and calculated their residual distance to each ancestor–descendant regression line. These residuals quantify the climatic deviation of each graph edge relative to the shifting niche optimum between ancestor and descendant. As species responses to deviations away from their climatic optima are usually modelled as quadratic rather than linear functions[117], linear distances were squared to produce the final climatic weights. Topographic weights were taken directly from the DEMs as the elevation-adjusted great circle distance between grid cell centres.

Each set of climatic and topographic weights was rescaled to the range [0–1], then multiplied together to produce the final set of graph weights. Similar compound weighting schemes have been used previously[104]. Each geographic connection therefore receives a cost of traversal optimized for the specific ancestor–descendant pathway under consideration: increasing distance from the minimal set of climatic conditions along the regression line in climate space results in a higher traversal cost, with additional penalization for topographically extreme routes. LCPs were identified between the spatiotemporal origination points of each pair of ancestor–descendant nodes across our four phylogenies using Dijkstra's algorithm. The geographic lengths of each LCP were calculated from the original topographic graph weights, then divided by the time-scaled branch lengths to quantify dispersal rates. Finally, we extracted the climatic conditions encountered along the LCPs from our BIOCLIM layers and projected those conditions into climate space.

### Ecographic diversity and dispersal rates

Our previous analyses yielded three sets of ancestral climatic tolerance estimates for each tree: (1) tolerances optimized from tip states using maximum likelihood estimation; (2) tolerances measured from geo model estimates of both tip and node locations; and (3) tolerances measured from geo model estimates and intervening TARDIS dispersal pathways. Each dataset was linearly interpolated at 0.1-Myr steps, then climate space occupancy (disparity) through time summarized from each set of ordinated tolerances. We used three metrics to, respectively, describe the volume, density and position of occupied climate space in all principal component axes: the sum of variances and ranges (extent of samples on all axes), average nearest neighbour distance (mean Euclidean distance between the pairwise distances of all samples) and average displacements (average position of samples relative to the dataset centroid). Datasets were binned at 2-Myr intervals and each bin bootstrapped 100 times to the original bin sample size. We calculated our chosen metrics from the bootstraps in each bin, taking the bin-wise means and 95% confidence intervals.

### Reporting summary

Further information on research design is available in the Nature Portfolio Reporting Summary linked to this article.

### Data availability

All data needed to replicate our analyses are available via Figshare at https://doi.org/10.6084/m9.figshare.28034708.v2 (ref. 73). The climate simulations used in this paper are archived on the BRIDGE research group server. All data needed to evaluate the conclusions in the paper are provided in ref. 73.

### Code availability

All scripts used to conduct our analyses are available via Figshare at https://doi.org/10.6084/m9.figshare.28034708.v2 (ref. 73). All software used is open source and available online. The TARDIS R package is available on GitHub (github.com/jf15558/TARDIS). The specific version used for the analyses here is included in ref. 73. PUMA codes for the simulations are also provided in ref. 73.

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

## Acknowledgements

This paper arose from a thesis submitted for fulfilment of a doctoral degree by J.T.F.-S. at the University of Bristol, funded by NERC GW4+DTP studentship S100065-138/123. We thank C. O'Donovan and A. Meade for assistance with the use of the BayesTraits geo model.

## Author contributions

A.E. constructed the phylogeny of early archosauromorphs and assisted J.T.F.-S. with further tree modifications and time-scaling. A.F. and D.J.L. ran the deep-time climate simulations. J.T.F.-S. conceived the study, developed the TARDIS spatiotemporal landscape model and conducted all other analysis. J.T.F.-S. wrote the manuscript with input and comments from A.E., A.F., D.J.L. and M.J.B.

## Competing interests

The authors declare no competing interests.

## Additional information

**Correspondence and requests for materials** should be addressed to Joseph T. Flannery-Sutherland.

# Reporting Summary

## Statistics

For all statistical analyses, confirm that the following items are present in the figure legend, table legend, main text, or Methods section.

| n/a | Confirmed | |
|---|---|---|
| ☐ | ☒ | The exact sample size (*n*) for each experimental group/condition, given as a discrete number and unit of measurement |
| ☐ | ☒ | A statement on whether measurements were taken from distinct samples or whether the same sample was measured repeatedly |
| ☐ | ☒ | The statistical test(s) used AND whether they are one- or two-sided<br>*Only common tests should be described solely by name; describe more complex techniques in the Methods section.* |
| ☒ | ☐ | A description of all covariates tested |
| ☒ | ☐ | A description of any assumptions or corrections, such as tests of normality and adjustment for multiple comparisons |
| ☐ | ☒ | A full description of the statistical parameters including central tendency (e.g. means) or other basic estimates (e.g. regression coefficient) AND variation (e.g. standard deviation) or associated estimates of uncertainty (e.g. confidence intervals) |
| ☒ | ☐ | For null hypothesis testing, the test statistic (e.g. $F$, $t$, $r$) with confidence intervals, effect sizes, degrees of freedom and $P$ value noted<br>*Give P values as exact values whenever suitable.* |
| ☐ | ☒ | For Bayesian analysis, information on the choice of priors and Markov chain Monte Carlo settings |
| ☒ | ☐ | For hierarchical and complex designs, identification of the appropriate level for tests and full reporting of outcomes |
| ☐ | ☒ | Estimates of effect sizes (e.g. Cohen's *d*, Pearson's *r*), indicating how they were calculated |

*Our web collection on statistics for biologists contains articles on many of the points above.*

## Software and code

Policy information about availability of computer code

| | |
|---|---|
| Data collection | Occurrence data were collated using the Paleobiology Database API via R scripts. Climate data was acquired and processed using R scripts from the University of Bristol climate simulation repository. |
| Data analysis | Two sets of analyses were conducted in MrBayes (tree timescaling; v.3.3.7) and BayesTraits (geo model estimation; v.4.0). All other analyses were conducted in R using a mixture of standard packages and custom functions. All code for all analyses is provided in the supplement |

For manuscripts utilizing custom algorithms or software that are central to the research but not yet described in published literature, software must be made available to editors and reviewers. We strongly encourage code deposition in a community repository (e.g. GitHub). See the Nature Portfolio guidelines for submitting code & software for further information.

## Data

Policy information about availability of data

All manuscripts must include a data availability statement. This statement should provide the following information, where applicable:
- Accession codes, unique identifiers, or web links for publicly available datasets
- A description of any restrictions on data availability
- For clinical datasets or third party data, please ensure that the statement adheres to our policy

All data is available in the repository link in the Data Availability statement in the manuscript.

# Research involving human participants, their data, or biological material

Policy information about studies with [human participants or human data](). See also policy information about [sex, gender (identity/presentation), and sexual orientation]() and [race, ethnicity and racism]().

| | |
|---|---|
| Reporting on sex and gender | NA |
| Reporting on race, ethnicity, or other socially relevant groupings | NA |
| Population characteristics | NA |
| Recruitment | NA |
| Ethics oversight | NA |

Note that full information on the approval of the study protocol must also be provided in the manuscript.

# Field-specific reporting

Please select the one below that is the best fit for your research. If you are not sure, read the appropriate sections before making your selection.

☐ Life sciences          ☐ Behavioural & social sciences          ☒ Ecological, evolutionary & environmental sciences

For a reference copy of the document with all sections, see [nature.com/documents/nr-reporting-summary-flat.pdf]()

# Ecological, evolutionary & environmental sciences study design

All studies must disclose on these points even when the disclosure is negative.

| | |
|---|---|
| Study description | Geographic and climatic niche ancestral estimation for early archosauromorph reptiles using a novel landscape connectivity approach |
| Research sample | A time-calibrated phylogeny and fossil occurrence data for 392 early archosauromorph taxa, ranging primarily from the Middle Permian to the Late Triassic. Climate simulations and topographic data for the same interval |
| Sampling strategy | All fossil occurrences were downloaded, then filtered in R |
| Data collection | Occurrences downloaded via the Palaeobiology Database using R |
| Timing and spatial scale | Fossil occurrence data ranging primarily from the Middle Permian to the Late Triassic, global scale dataset |
| Data exclusions | Only a few occurrences were excluded as taxonomically suspect. All exclusions are fully documented in the ESM |
| Reproducibility | Every single analysis and data processing protocol is provided in the ESM using R, bash, MrBayes and BayesTraits scripts |
| Randomization | Not relevant |
| Blinding | Not relevant |

Did the study involve field work?     ☐ Yes     ☒ No

# Reporting for specific materials, systems and methods

We require information from authors about some types of materials, experimental systems and methods used in many studies. Here, indicate whether each material, system or method listed is relevant to your study. If you are not sure if a list item applies to your research, read the appropriate section before selecting a response.

## Materials & experimental systems

| n/a | Involved in the study |
|---|---|
| ☒ | Antibodies |
| ☒ | Eukaryotic cell lines |
| ☒ | Palaeontology and archaeology |
| ☒ | Animals and other organisms |
| ☒ | Clinical data |
| ☒ | Dual use research of concern |
| ☒ | Plants |

## Methods

| n/a | Involved in the study |
|---|---|
| ☒ | ChIP-seq |
| ☒ | Flow cytometry |
| ☒ | MRI-based neuroimaging |

## Plants

| Seed stocks | NA |
|---|---|
| Novel plant genotypes | NA |
| Authentication | NA |

