## [Peer Review File · Nature Ecology & Evolution]

Landscape-explicit phylogeography illuminates the ecographic radiation of early archosauromorph reptiles

Corresponding Author: Dr Joseph Flannery Sutherland

Version 0:

Decision Letter:

29th January 2025

Dear Dr Flannery Sutherland,

Your manuscript entitled "Landscape-explicit phylogeography illuminates the ecographic radiation of early archosauromorph reptiles" has now been seen by three reviewers, whose comments are attached. The reviewers have raised a number of concerns which will need to be addressed before we can offer publication in Nature Ecology & Evolution. We will therefore need to see your responses to the criticisms raised and to some editorial concerns, along with a revised manuscript, before we can reach a final decision regarding publication.

Please pay particular attention to reviewer 1's comments regarding incomplete fossil record sampling, and reviewer 3's regarding sampling alternative topologies.

We therefore invite you to revise your manuscript taking into account all reviewer and editor comments. Please highlight all changes in the manuscript text file [OPTIONAL: in Microsoft Word format].

* If you have not done so already please begin to revise your manuscript so that it conforms to our Article format instructions at <http://www.nature.com/natecolevol/info/final-submission>. Refer also to any guidelines provided in this letter.

* Extended Data Figures - please ensure that any supplementary figures and tables that are crucial to the manuscript's conclusions are converted into Extended Data figures and tables to increase visibility of these data. Extended Data figures and tables are online-only (present in the online PDF and full-text HTML versions of the paper), peer-reviewed display items that provide essential background to the article but are not included in the main article due to space constraints. A maximum of ten Extended Data display items (figures and tables) is permitted.

Link Redacted

Nature Ecology & Evolution is committed to improving transparency in authorship. As part of our efforts in this direction, we are now requesting that all authors identified as 'corresponding author' on published papers create and link their Open Researcher and Contributor Identifier (ORCID) with their account on the Manuscript Tracking System (MTS), prior to acceptance. ORCID helps the scientific community achieve unambiguous attribution of all scholarly contributions. You can create and link your ORCID from the home page of the MTS by clicking on 'Modify my Springer Nature account'. For more information please visit www.springernature.com/orcid.

[redacted]

Reviewer expertise:

Reviewer #1: signed review

Reviewer #2: phylogeography, ecological niche modelling, extant species

Reviewer #3: signed review

Reviewers' comments:

Reviewer #1 (Remarks to the Author):

Overview

This is an interesting paper that is important not only for its contribution to our understanding of the early evolution of archosauromorphs, but also because it presents a new analytical method for looking at biogeography, fossil record sampling, and climatic niche occupancy in deep time. I found the paper easy to follow (though see my specific comments below regarding some aspects of clarity and terminology). I would particularly like to complement the authors on the excellence of the discussion section which is very explicit and clear about the limitations and caveats that apply to the study, especially with regard to fossil record sampling. One could certainly argue about the validity of certain assumptions underpinning this work, and no doubt some other reviewers may choose to do so. However, any modelling approach of this type must inevitably balance realism against pragmatism. Moreover, this clearly represents an advanced in an interesting direction and is something that I believe other researchers will want to see and use. I am therefore happy to support publication of this work subject to the authors addressing the issues I have set out below.

Specific points

One of the conclusions from this paper is to support the long-held view that dinosaurs originated in South America. However, as the authors note themselves, there are concerns regarding incomplete fossil record sampling, and this may well have affected our views on dinosaur origins. In particular, a recent paper which came out while the current study was in review (Lovelace et al 2025 Rethinking dinosaur origins... Zoological Journal of the Linnean society), reports an interesting low palaeolatitude fauna from the late Carnian of North America. I am not asking the authors to re-run any of their analyses, but I think they should acknowledge the existence of this paper and compare its implications with their inferences – this might help illustrate some of the limitations and caveats they note in their discussion section.

Lines 66-67 – "(Dunne et al., 2021, 2023). These narratives, however, are based on inferences from their sporadically sampled fossil record and basal phylogenetic relationships. Further, while early..."

I recommend changing "basal" to "early diverging" because of the problems associated with "basal" in a phylogenetic context – see Omland et al. -

Omland, K. E., L. G. Cook, and M. D. Crisp. 2008. Tree thinking for all biology: the problem with reading phylogenies as ladders of progress. *BioEssays* 30:854–867.

Also, "Narratives" is potentially problematic and I would recommend changing this to something like "scenarios" or even "hypotheses". "Narratives" in the context of biogeography has come to mean an account based largely on a literal reading of the fossil record (see Ball 1975 and papers that have cited this subsequently). However, the various studies cited at this point in the paper include many that have made decent attempts to take sampling into account and fill in gaps using a variety of techniques including those based on a phylogenetic framework. The authors may well think that these techniques are flawed in some way or could be improved upon, but I think it will muddy the waters somewhat if they describe the results as "narratives".

Ball, I. R. 1975. Nature and formulation of biogeographic hypotheses. *Systematic Zoology*, 24, 407–430.

Lines 106-114 - "...in western Gondwana). Originations within the archosauromorph crown group are sensitive to the placement of traditional stem dinosaur clades (Fig. 2). Placement of lagerpetids as the sister group of pterosaurs (Ezcurra et al., 2020), and silesaurids as a basal grade of ornithischian dinosaurs (Muller and Garcia 2020) collapses poorly resolved bimodal estimates for the geographic origins of archosaurs and various pseudosuchian clades to more robust unimodal estimates (Fig. 2A, B, Supplementary Data 1), and substantially reduces divergence time uncertainty for pterosauroforms, pterosaurs and ornithischians (Fig. 2C, D). All subsequent results presented here are based on a topology with revised positions of both silesaurids and ornithischians".

Again, avoid "basal" in the context of phylogenetic relationships.

Also, I think the last line should read "All subsequent results presented here are based on a topology with revised positions of both silesaurids and lagerpetids". As you can see, the previous sentences in lines 106-114 talk about two aspects of topology – the positions of silesaurids and lagerpetids, but do not talk about the position of ornithischians. So the final line as originally written is a bit confusing because there has been no previous reference to the position of Ornithischia and I can only make sense of it if I substitute "lagerpetids" for "ornithischians".

Lines 474-476 – "...of highest density cell in the grid, and the 95% highest density intervals for the estimated longitudes and latitudes."

Here, and at several other points, I think that "latitude" and "longitude" should actually be "palaeolatitude" and "Palaeolongitude", respectively. Each fossil specimen has a modern day latitude and longitude where it was found, and an almost certainly different Palaeolatitude and palaeolongitude where the living animal existed. If we are talking about estimating the location of an ancestor, for example, then this should be expressed as palaeolatitude/Palaeolongitude in order to keep this distinct from any confusion with the location of fossils today. Not only is this important when describing evolutionary scenarios in deep time, but also in any study that touches on sampling biases. Sampling bias may relate to events that happened at a particular place in the past (e.g. geological factors such as sedimentation and erosion - in which case discussing their palaeo location is relevant), or they may relate to modern day biases (e.g. those caused by colonialism, other anthropogenic biases, difficulty in working in certain harsh environments and So on – in which case the terms latitude and longitude would help keep it clear that we were talking about the present time). So, I think the authors need to go through the manuscript carefully and ensure that they use the appropriate terminology depending on the context.

Paul Upchurch

Reviewer #2 (Remarks to the Author):

This is an ambitious paper that sets out to track the spatial origins and biogeographic movements of the earliest archosauromorphs through the use of phylogenetic niche modeling and a newly developed spatiotemporal phylogeographic path analysis method. The paper was somewhat difficult for me as my scientific expertise is in phylogeography and niche modeling, while my knowledge of paleontology is more that of an interested layman (for instance, I have the major geological time periods memorized, but not the smaller stages within them). Nonetheless, I was overall quite impressed by the paper and its methodology. My critique is fairly limited. Mostly I wish the authors to better integrate certain topics into the discussion, and to simplify the wording of certain ideas to make them not just easier to understand, but make it easier to see how they affect the big picture and the results overall. For instance, in Figure 6, it is not entirely clear from the text what the biological and ecological meanings of the displayed statistics are, and how the differences between the ways they were calculated (i.e., the three different colored lines in each plot) matter (again, in a bio/ecological context). This is all probably clear to the author(s), but in my first pass through (which is the very most you can hope for from most readers), these nuances didn't really work their way into my brain. I would also like to see a deeper discussion of caveats and issues related to the phylogenetic niche modeling part of the paper. Overall, I commend the authors on their ambitious undertaking and appreciate being sent this paper that combines a professional interest (niche modeling) with a casual, more fanboy-ish one (dinosaurs!).

Major comments:

The discussion of caveats relating to potential sampling bias and the geo model was well done, but I was hoping for more discussion of the caveats of the niche modeling method itself, both in terms of the climate and occurrence data used, and the modeling methodology. Many readers such as myself that come from a niche modeler/phylogeography background vs. a paleontological one will desire a deeper discussion of this topic.

Lines 539-541: Here it is stated that nodal climate conditions were estimated with ML using "extracted climatic conditions for the fossil tips." What exactly does this mean? Specifically, I would be interested to know what the actual values being used in the ancestral state reconstruction are. I think this should be stated more clearly for the reader overall.

I wish the differences between climate space occupancy derived from tips & ML estimates, tips & ancestral locations, and

ancestral locations & dispersal paths was explained better prior to or early on in the Results, both in terms of how these values are calculated and what they mean biologically compared to each other.

Minor comments:

Lines 174-185: I recommend continuing to provide references to Fig 5 throughout this paragraph (as with line 177) to make it easier for the reader to follow along.

Line 247: I'm not sure what "once geographically proximate" means here.

Lines 880-881: I believe this should be referring to Figure 3.

Line 919: I believe this should refer to Supplementary Data 3.

I recommend making font sizes larger in many figures, such as Fig. 3. Many of these labels are too small to read at normal sizes.

Reviewer #3 (Remarks to the Author):

The authors present a novel approach to reconstructing the biogeographic history of long extinct lineages. By utilizing the geographic and temporal locations of species within a greater clade, the authors estimate clade-wide dispersal maps for each stage from the end of the Early Permian (Kungurian) to the end of the Triassic (Rhaetian), based on recent phylogenies. This work is interesting and provides a unique perspective on reconstructing ancient biogeographic hypotheses at an unprecedented level of detail. This work should certainly be published, and I believe that *Nature Ecology & Evolution* would be a strong fit for this work. With that being said, I believe that the underlying phylogeny may not be the best to address this question. The topology presented is based off of an analysis from 2017 that has had numerous modifications to it in the intervening years, some of which include topologies that are contradictory to the narrative presented herein. Most notable of which are the Lagerpetid avemetatarsalians which are represented as having an orthodox placement as a monophyletic clade among the dinosaurian stem. The authors represent a heterodox relationship for Lagerpetids as the sister taxon to Pterosauria. However, there have been a number of papers within the last few years that have presented Lagerpetids as the sister taxon to Pterosauria, with increasing confidence in each publication. I think it would benefit the others to reframe their argument to match the currently accepted topologies for these groups, some of which have been published quite recently. Additionally, there are existing arguments that suggest that bayesian tip-dated phylogenies are more appropriate than time-scaled parsimonious trees, as the bayesian estimations tend to more closely match the given stratigraphic records. Given that there have been a series of recent publications that have performed both parsimony and tip-dated bayesian inference on archosauromorphs, dinosaurs, and crocodiles, I think it would likely benefit the authors to incorporate a tip-dated bayesian supertree with the most contemporary topologies either alongside or in lieu of the existing topology. I understand this is a big ask, but I think doing so has the potential to provide even more clarity and resolution to this work.

If you have any questions regarding my review, I would be happy to chat with you more in depth.

Best,
Brenen Wynd

Remarks to the author:

Major comments:

1. I believe that there are some core issues to the phylogenetic materials chosen to base the supertree off of that need to be addressed. The phylogeny presented includes a number of taxa that are generally removed from the trees a priori because of their problematic nature of being highly incomplete (*Saltopus* is known from a mould) or potentially chimaeric (*Agnosphytis*), and so I would recommend a large portion of the tree be removed for this work. Ezcurra & Sues (2022) represents an updated iteration of this tree, and in their phylogenetic methods they list 40 taxa that were pruned from the analysis a priori as they are only used for disparity analyses. Many of these taxa are on the phylogeny figured, and so the phylogeny does bear a significant portion of taxa whose actual phylogenetic position is extremely challenging to place due to their incomplete nature. Additionally, Griffin et al., (2022) includes personal observations from Sterling Nesbitt (who described *Nyasasaurus*) whom attests that the materials of both *Agnosphytis* and *Nyasasaurus* are unlikely to reflect only two unique species.

Personally, I do not agree with using a time-scaling approach on a tree derived from equal-weights parsimony. Parsimony lacks inherently informative branch lengths, and it has been shown that a bayesian tip-dating analysis produces results more consistent with stratigraphic reality than time-scaling (King 2021). I suggest that you build your supertree based on the results of tip-dated phylogenies from bayesian inference. I have included references to tip-dated phylogenies published for early archosaurs and non-dinosaurian dinosauriformes (Muller et al 2023), dinosaurs (Griffin et al 2022), and pseudosuchians (Turner et al. 2017). Generating a tip-dated supertree from these results (using Muller et al 2023 as the core of the tree), would provide more informative branch lengths and may impact the duration of the estimated dispersal events.

2. Framing the traditional tree as having Lagerpetids and Silesaurids as monophyletic stem dinosaur clades is in conflict with the current accepted hypotheses surrounding both of these groups. The 'control' topology for which everything is tested against should have Silesaurids as a paraphyletic grade just outside of Dinosauria, and also should have Lagerpetids as an early-diverging pterosauriforms that are sister to Pterosauria, as is presented in Muller et al (2023). I think it is fine to keep

the comparison where both clades are considered stem dinosaurs, but framing that as the contemporary hypothesis is a bit outdated.

Minor comments:

Line 41: change to "if fossil data are"

Line 60: I recommend changing instances of "basal" to "early-diverging". The term basal implies some degree of primitiveness, but many of these earliest diverging taxa lived within five million years of one another, which would suggest that they had evolved for roughly the same amount of time. I understand if you do not wish to make this change, I just wanted to make a case for what I consider to be a more accurate term, and so I will not comment on any other instances of this.

Line 67: change "while" to "although"

Line 114: Is this meant to be silesaurids and ornithischians or silesaurids and lagerpetids?

Line 146: There is evidence that the Doswelliidae is a northern radiation sister to Proterochampsia. Bayesian analysis by Wynd et al (2019) and implied weights parsimony analyses done by Ezcurra and Sues (2022) both recover the Doswelliidae as being sister to the genus Proterochampsia, the earliest diverging of the proterochampsid genera. Because implied weights and bayesian inference with gamma distributed rate categories are adjusting the impacts that characters have on the overall topology, they are both essentially treating characters as if they evolve at different rates. Because it is highly unlikely that the hundreds of traits in the phylogenetic analyses would all share the exact same rate and phyletic history, I argue that the models with rate heterogeneity are likely more accurate. However, Muller et al (2023) do not recover the same relationship in their analysis, though that may be in part because they were focusing more of their attention elsewhere on the tree. It may be interesting to include this as another heterodox topology to test your model against, simply because I would imagine that embedding a Laurasian radiation near the base of an otherwise gondwanan clade may require a more intermediate ancestral location and thus unique dispersal patterns. I only suggest this because the Eucrocopoda are currently represented as the sister taxon to Archosauria and so a shift in the ancestral location for that clade may also impact reconstructions for the earliest diverging archosaur species.

Lines 255-258: I disagree with this statement given the fact that you cannot perform a molecular analysis on these samples, and so you are bound to what you have access to. Additionally, the reference given for Oyston simply frames the relationship as morphological vs molecular whereas the reality of paleontology is a question of likelihood (trending bayesian) and parsimony. King (2021) suggests that bayesian tip-dated trees are more stratigraphically congruent and thus may be a better framework for your kind of question.

Line 301: The acronym EPME is only used twice throughout the manuscript. I would writing the entire phrase if only to prevent your reader from needing to go back to remember what it stood for in the introduction.

Line 384: I might suggest citing someone more contemporary than Seeley. Perhaps an early career scientist where citations could have a beneficial impact on a budding career.

Fig 1) The first fossil appearance for Allokotosauria is deep in the Norian, I'm assuming affiliated with Trilophosauridae? This should be back in the anisian based on Shringasaurus (Sengupta et al 2017).

Labels for which time periods the paleomaps are based off of would be helpful to better gauge exactly when these plots are in reference to.

"Crown Ornithischia" should just be "Ornithischia." A crown group refers to the least inclusive clade that includes all presently living members of the group, and so Ornithischia cannot be considered a crown group.

References

Ezcurra, Martín D., and Hans-Dieter Sues. "A re-assessment of the osteology and phylogenetic relationships of the enigmatic, large-headed reptile *Sphodrosaurus pennsylvanicus* (Late Triassic, Pennsylvania, USA) indicates archosauriform affinities." *Journal of Systematic Palaeontology* 19, no. 24 (2021): 1643-1677.

Ezcurra, Martín D., Lucas E. Fiorelli, Agustín G. Martinelli, Sebastián Rocher, M. Belén von Baczko, Miguel Ezpeleta, Jeremías RA Taborda, E. Martín Hechenleitner, M. Jimena Trotteyn, and Julia B. Desojo. "Deep faunistic turnovers preceded the rise of dinosaurs in southwestern Pangaea." *Nature Ecology & Evolution* 1, no. 10 (2017): 1477-1483.

Griffin, Christopher T., Brenen M. Wynd, Darlington Munyikwa, Tim J. Broderick, Michel Zondo, Stephen Tolan, Max C. Langer, Sterling J. Nesbitt, and Hazel R. Taruvinga. "Africa's oldest dinosaurs reveal early suppression of dinosaur distribution." *Nature* 609, no. 7926 (2022): 313-319.

King, Benedict. "Bayesian tip-dated phylogenetics in paleontology: topological effects and stratigraphic fit." *Systematic Biology* 70, no. 2 (2021): 283-294.

Müller, Rodrigo T., Martín D. Ezcurra, Mauricio S. Garcia, Federico L. Agnolín, Michelle R. Stocker, Fernando E. Novas, Marina B. Soares, Alexander WA Kellner, and Sterling J. Nesbitt. "New reptile shows dinosaurs and pterosaurs evolved among diverse precursors." *Nature* 620, no. 7974 (2023): 589-594.

Sengupta, Saradee, Martín D. Ezcurra, and Saswati Bandyopadhyay. "A new horned and long-necked herbivorous stem-archosaur from the Middle Triassic of India." *Scientific reports* 7, no. 1 (2017): 8366.

Turner, Alan H., Adam C. Pritchard, and Nicholas J. Matzke. "Empirical and Bayesian approaches to fossil-only divergence times: a study across three reptile clades." *PloS one* 12, no. 2 (2017): e0169885.

Wynd, Brenan M., Sterling J. Nesbitt, Michelle R. Stocker, and Andrew B. Heckert. "A detailed description of *Rugarhynchos sixmilensis*, gen. et comb. nov. (Archosauriformes, Proterochampsia), and cranial convergence in snout elongation across stem and crown archosaurs." *Journal of Vertebrate Paleontology* 39, no. 6 (2019): e1748042.

*****END*****

Version 1:

Decision Letter:

7th March 2025

Dear Joe

Thank you for submitting your revised manuscript "Landscape-explicit phylogeography illuminates the ecographic radiation of early archosauromorph reptiles" (NATECOLEVOL-24113384A). It has now been seen again by the original reviewers and their comments are below. The reviewers find that the paper has improved in revision, and therefore we'll be happy in principle to publish it in *Nature Ecology & Evolution*, pending minor revisions to satisfy the reviewers' final requests and to comply with our editorial and formatting guidelines.

Thank you again for your interest in *Nature Ecology & Evolution*. Please do not hesitate to contact me if you have any questions.

[redacted]

Reviewer #1 (Remarks to the Author):

Review of revised version

I have been through the revised manuscript and the rebuttal letter. I note that the authors have made all of the changes I requested. Reviewers 3 and 4 raise issues about the supertree used (Bayesian versus Parsimony), and advocate for the use of Bayesian trees which in turn would have necessitated rerunning all analyses. I completely agree with the authors that this is something they should not be compelled to do. Aside from the pragmatic issues of obtaining computing time noted by the authors, I agree with the suggestion that such a major change is unlikely to radically alter their conclusions. Moreover, for me, the importance of this paper lies more in the methodological novelty (which will have a strong impact on the field by providing a new set of techniques for the rest of us to play with), rather than the specifics of archosauromorph biogeography. There are certainly some newsworthy and interesting aspects to what the paper says about archosauromorphs, but fundamentally the results are not too surprising in that regard – it is the methodological novelty that makes this paper particularly important and why I believe it should be published. Therefore, based on the revised manuscript and the arguments made in the rebuttal letter, I would now support acceptance of the paper in its current form.

Reviewer #2 (Remarks to the Author):

Thank you to the authors for addressing my few comments, which mostly addressed a few topics I wanted additional clarity on.

L325-348: I thought the new paragraph added to the discussion about the limitations of the niche modeling approach was well done and addressed my desire for proper discussion of this topic.

L562-563: Thank you for clarifying what the source of the climatic tip states is.

L178-187: The new explanation of the various ways you quantified archosauromorph niches is helpful, thank you.

The new version of the manuscript is ready to go. I commend the authors on their work.

Reviewer #2 (Remarks on code availability):

n/a

Reviewer #3 (Remarks to the Author):

The authors took the critiques of myself and the other reviewers and offered well-reasoned responses in cases where comments weren't feasible to be incorporated. I think the authors responses and associated corrections made to the manuscript warrant publication in Nature Ecology & Evolution. The inclusion of SCI and GER tree statistics provides a useful justification for the choice of tree used herein. Additionally, the language surrounding phylogenetic and systematic relationships are more clear.

Response to the response:

"Secondly, we take conceptual issue with post-hoc timescaling of a tree that had already incorporated stratigraphic information from fossils into its topology."

Apologies for my poor phrasing, I was not intending to recommend timescaling a bayesian tree. My intent was that the construction of a supertree from established bayesian topologies would remove the need for post-hoc scaling. I agree that time-scaling a bayesian topology would be both redundant and counter-productive.

Minor comment:

Line 402: Change to "We also elected to retain the interrelationships of eucoelopodans and early-diverging pseudosuchians in our source tree,..."

Doswelliid arguments for being psuedosuchian are not supported today. Including Eucoelopoda better encapsulates relationships within and outside of Archosauria.

Reviewer #3 (Remarks on code availability):

I include comments regarding the code availability in confidential remarks to the editor. But as far as I can tell (and I tried searching the websites mentioned in the Data and Code availability sections), the data and code are not presently available or made public to anyone but the authors. If this is not the case, then there are not appropriate links or avenues to reasonably and easily find the supplementary information for this project.

Response to Reviewers

Reviewer comments are in black. Our responses are in red. All changes in response to reviewer comments are referenced by line number in this document and highlighted in the revised manuscript file in yellow

Reviewer #1 (Remarks to the Author):

Overview

This is an interesting paper that is important not only for its contribution to our understanding of the early evolution of archosauromorphs, but also because it presents a new analytical method for looking at biogeography, fossil record sampling, and climatic niche occupancy in deep time. I found the paper easy to follow (though see my specific comments below regarding some aspects of clarity and terminology). I would particularly like to complement the authors on the excellence of the discussion section which is very explicit and clear about the limitations and caveats that apply to the study, especially with regard to fossil record sampling. One could certainly argue about the validity of certain assumptions underpinning this work, and no doubt some other reviewers may choose to do so. However, any modelling approach of this type must inevitably balance realism against pragmatism. Moreover, this clearly represents an advanced in an interesting direction and is something that I believe other researchers will want to see and use. I am therefore happy to support publication of this work subject to the authors addressing the issues I have set out below.

We thank the reviewer for their positive outlook on the paper and have made every effort to incorporate their feedback into the revised manuscript.

Specific points

One of the conclusions from this paper is to support the long-held view that dinosaurs originated in South America. However, as the authors note themselves, there are concerns regarding incomplete fossil record sampling, and this may well have affected our views on dinosaur origins. In particular, a recent paper which came out while the current study was in review (Lovelace et al 2025 Rethinking dinosaur origins... Zoological Journal of the Linnean society), reports an interesting low palaeolatitude fauna from the late Carnian of North America. I am not asking the authors to re-run any of their analyses, but I think they should acknowledge the existence of this paper and compare its implications with their inferences – this might help illustrate some of the limitations and caveats they note in their discussion section.

We are grateful to the reviewer for drawing our attention to this paper. We have incorporated it into our discussion (L252-256), along with another paper out around the same time that explicitly models a lower latitude origin for dinosaurs, and believe that the alternative results they recover strengthen our message that biogeographic origins are heavily contingent on sampling in the fossil record.

Lines 66-67 – “(Dunne et al., 2021, 2023). These narratives, however, are based on inferences from their sporadically sampled fossil record and basal phylogenetic relationships. Further, while early...”.

I recommend changing “basal” to “early diverging” because of the problems associated with “basal” in a phylogenetic context – see Omland et al. (2008)

Omland, K. E., L. G. Cook, and M. D. Crisp. 2008. Tree thinking for all biology: the problem with reading phylogenies as ladders of progress. *BioEssays* 30:854–867.

Change made to ‘early diverging’ here and throughout the rest of the manuscript (L55, 62, 92, 101, 110, 123, 138, 353, 399, 407, 1178, 1187)

Also, “Narratives” is potentially problematic and I would recommend changing this to something like “scenarios” or even “hypotheses”. “Narratives” in the context of biogeography has come to mean an account based largely on a literal reading of the fossil record (see Ball 1975 and papers that have cited this subsequently). However, the various studies cited at this point in the paper include many that have made decent attempts to take sampling into account and fill in gaps using a variety of techniques including those based on a phylogenetic framework. The authors may well think that these techniques are flawed in some way or could be improved upon, but I think it will muddy the waters somewhat if they describe the results as “narratives”.

Ball, I. R. 1975. Nature and formulation of biogeographic hypotheses. *Systematic Zoology*, 24, 407–430.

Change made to ‘scenarios’ (L61)

Lines 106-114 - “...in western Gondwana). Originations within the archosauromorph crown group are sensitive to the placement of traditional stem dinosaur clades (Fig. 2). Placement of lagerpetids as the sister group of pterosaurs (Ezcurra et al., 2020), and silesaurids as a basal grade of ornithischian dinosaurs (Muller and Garcia 2020) collapses poorly resolved bimodal estimates for the geographic origins of archosaurs and various pseudosuchian clades to more robust unimodal estimates (Fig. 2A, B, Supplementary Data 1), and substantially reduces divergence time uncertainty for pterosauromorphs, pterosaurs and ornithischians (Fig. 2C, D). All subsequent results presented here are based on a topology with revised positions of both silesaurids and

ornithischians”.

Again, avoid “basal” in the context of phylogenetic relationships.

Addressed as above

Also, I think the last line should read “All subsequent results presented here are based on a topology with revised positions of both silesaurids and lagerpetids”. As you can see, the previous sentences in lines 106-114 talk about two aspects of topology – the positions of silesaurids and lagerpetids, but do not talk about the position of ornithischians. So the final line as originally written is a bit confusing because there has been no previous reference to the position of Ornithischia and I can only make sense of it if I substitute “lagerpetids” for “ornithischians”.

This was a typo on our part and the change has been made from ‘ornithischians’ to ‘lagerpetids’ (L107)

Lines 474-476 – “...of highest density cell in the grid, and the 95% highest density intervals for the estimated longitudes and latitudes.”

Here, and at several other points, I think that “latitude” and “longitude” should actually be “palaeolatitude” and “Palaeolongitude”, respectively. Each fossil specimen has a modern day latitude and longitude where it was found, and an almost certainly different Palaeolatitude and palaeolongitude where the living animal existed. If we are talking about estimating the location of an ancestor, for example, then this should be expressed as palaeolatitude/Palaeolongitude in order to keep this distinct from any confusion with the location of fossils today. Not only is this important when describing evolutionary scenarios in deep time, but also in any study that touches on sampling biases. Sampling bias may relate to events that happened at a particular place in the past (e.g. geological factors such as sedimentation and erosion - in which case discussing their palaeo location is relevant), or they may relate to modern day biases (e.g. those caused by colonialism, other anthropogenic biases, difficulty in working in certain harsh environments and So on – in which case the terms latitude and longitude would help keep it clear that we were talking about the present time). So, I think the authors need to go through the manuscript carefully and ensure that they use the appropriate terminology depending on the context.

We thank the reviewer for highlighting this potential source of confusion for readers and agree with their point. We have gone through the manuscript and implemented these changes throughout (L253, 448, 498, 504, 507, 515, 554, 646)

Paul Upchurch

Reviewer #2 (Remarks to the Author):

This is an ambitious paper that sets out to track the spatial origins and biogeographic movements of the earliest archosauromorphs through the use of phylogenetic niche modeling and a newly developed spatiotemporal phylogeographic path analysis method. The paper was somewhat difficult for me as my scientific expertise is in phylogeography and niche modeling, while my knowledge of paleontology is more that of an interested layman (for instance, I have the major geological time periods memorized, but not the smaller stages within them). Nonetheless, I was overall quite impressed by the paper and its methodology. My critique is fairly limited. Mostly I wish the authors to better integrate certain topics into the discussion, and to simplify the wording of certain ideas to make them not just easier to understand but make it easier to see how they affect the big picture and the results overall. For instance, in Figure 6, it is not entirely clearly from the text what the biological and ecological meanings of the displayed statistics are, and how the differences between the ways they were calculated (i.e., the three different colored lines in each plot) matter (again, in a bio/ecological context). This is all probably clear to the author(s), but in my first pass through (which is the very most you can hope for from most readers), these nuances didn't really work their way into my brain. I would also like to see a deeper discussion of caveats and issues related to the phylogenetic niche modeling part of the paper. Overall, I commend the authors on their ambitious undertaking and appreciate being sent this paper that combines a professional interest (niche modeling) with a casual, more fanboy-ish one (dinosaurs!).

We thank the reviewer for their positive response to the manuscript. Following their feedback we have carefully checked our phrasing throughout the manuscript to ensure that it is accessible as possible for readers across different disciplines.

Major comments:

The discussion of caveats relating to potential sampling bias and the geo model was well done, but I was hoping for more discussion of the caveats of the niche modeling method itself, both in terms of the climate and occurrence data used, and the modeling methodology. Many readers such as myself that come from a niche modeler/phylogeography background vs. a paleontological one will desire a deeper discussion of this topic.

We have added a new paragraph to our discussion to give a more thorough treatment of the limitations of our climate simulation and occurrence data with regards to niche modelling and dispersal pathway estimation (L325-348)

Lines 539-541: Here it is stated that nodal climate conditions were estimated with ML using "extracted climatic conditions for the fossil tips." What exactly does this mean? Specifically, I would be interested to know what the actual values being used in the ancestral state reconstruction are. I think this should be stated more clearly for the reader overall.

We have rephrased the text to make the source of the climatic tip states absolutely clear (L562-563).

I wish the differences between climate space occupancy derived from tips & ML estimates, tips & ancestral locations, and ancestral locations & dispersal paths was explained better prior to or early on in the Results, both in terms of how these values are calculated and what they mean biologically compared to each other.

We have revised our presentation of the climatic disparity results so that the biological meanings of our summary statistics are clearly defined prior to describing any trends (L178-187). Specifically, we now refer to the diversity and consistency of occupied climatic niches based on the range and density of dispersal pathways in climate space. We have also reworked the caption for Fig. 6 so that these meanings are also conveyed to the figure, rather than requiring the reader to flick between text and figure (L1225-1234)

Minor comments:

Lines 174-185: I recommend continuing to provide references to Fig 5 throughout this paragraph (as with line 177) to make it easier for the reader to follow along.

Additional references to Fig. 5 have been added throughout the paragraph (L171, 174, 177)

Line 247: I'm not sure what "once geographically proximate" means here.

We have changed 'proximate' to 'closer' to keep our meaning clear (L241)

Lines 880-881: I believe this should be referring to Figure 3.

A typo on our part. Corrected to Fig. 3 (L166)

Line 919: I believe this should refer to Supplementary Data 3.

This was another typo. We have elected to convert this particular file to an Extended Data Figure and have updated the text accordingly (L141, 1206)

I recommend making font sizes larger in many figures, such as Fig. 3. Many of these labels are too small to read at normal sizes.

We agree that our labels were quite small in many cases. Where possible we have enlarged the text to address these issues, although in some cases this was not always possible (notably the tip labels for the phylogeny). With regards to this specific instance, this is also why we thought it prudent to present a simplified version of the phylogeny in Fig. 1. This way a reader can still get a clear sense of the taxonomic scope of the tree, then refer to Fig. 3 in the online version for the specific taxa included.

Reviewer #3 (Remarks to the Author):

The authors present a novel approach to reconstructing the biogeographic history of long extinct lineages. By utilizing the geographic and temporal locations of species within a greater clade, the authors estimate clade-wide dispersal maps for each stage from the end of the Early Permian (Kungurian) to the end of the Triassic (Rhaetian), based on recent phylogenies. This work is interesting and provides a unique perspective on reconstructing ancient biogeographic hypotheses at an unprecedented level of detail. This work should certainly be published, and I believe that *Nature Ecology & Evolution* would be a strong fit for this work. With that being said, I believe that the underlying phylogeny may not be the best to address this question. The topology presented is based off of an analysis from 2017 that has had numerous modifications to it in the intervening years, some of which include topologies that are contradictory to the narrative presented herein. Most notable of which are the Lagerpetid avemetatarsalians which are represented as having an orthodox placement as a monophyletic clade among the dinosaurian stem. The authors represent a heterodox relationship for Lagerpetids as the sister taxon to Pterosauria. However, there have been a number of papers within the last few years that have presented Lagerpetids as the sister taxon to Pterosauria, with increasing confidence in each publication. I think it would benefit the others to reframe their argument to match the currently accepted topologies for these groups, some of which have been published quite recently. Additionally, there are existing arguments that suggest that bayesian tip-dated phylogenies are more appropriate than time-scaled parsimonious trees, as the bayesian estimations tend to more closely match the given stratigraphic records. Given that there have been a series of recent publications that have performed both parsimony and tip-dated bayesian inference on archosauromorphs, dinosaurs, and crocodiles, I think it would likely benefit the authors to incorporate a tip-dated bayesian supertree with the most contemporary topologies either alongside or in lieu of the

existing topology. I understand this is a big ask, but I think doing so has the potential to provide even more clarity and resolution to this work.

We are grateful to the reviewer for their feedback on our manuscript and appreciate the concerns raised by advances made in time-scaling methodologies and early archosauromorph topologies. The reviewer's points have given us a valuable opportunity to reflect on the limitations of our work and as such we have given them greater emphasis in our discussion to ensure that the phylogenetic caveats are abundantly clear (L252-256), particularly the potential for divergent biogeographic results as demonstrated by very recently published work on early dinosaurian origins (Heath et al., 2025).

The critical issue is whether a revised tree would make a significant practical difference to the results and conclusions we currently present. While we agree in principle that Bayesian tip-dated methods may be preferable to parsimony with post-hoc time-scaling, we do not think that the substantial task of re-running out analyses with a revised tree would incur significant gain in terms of either the overall accuracy of our findings or the implications we draw from them. This partly stems from our assertion that the differences of taxon sampling, topology and branch lengths will be small enough so as not to alter message of our paper.

Ultimately, our goal is to present a novel approach to tackle the issue of incompletely sampled niches in the fossil record, using tree wide results rather than the dynamics of specific subclades. Where we do discuss the biogeographic origins of archosauromorphs, this is mainly to provide a scaffold for our subsequent investigation of Hutchinson's duality across the entire tree. In turn this is why we continue to carefully highlight the sensitivity of any biogeographic analysis to topological choice and new fossil discoveries, rather than presenting our specific findings as an incontrovertible reconstruction of their past biogeography.

Given the undeniable importance of methodological and topological debate the reviewer points out, we have provided an extensive and thorough reasoning below to justify our adherence to our current topology and methods. In addition, we would also face major difficulties in rerunning any analyses due to unavailability of the required computational resources, as a secondary reason for retaining the results presented herein.

If you have any questions regarding my review, I would be happy to chat with you more in depth.

Best,
Brenen Wynd

Remarks to the author:

Major comments:

1. I believe that there are some core issues to the phylogenetic materials chosen to base the supertree off of that need to be addressed.

Ultimately, we do not believe that re-analysis using an alternative supertree would strongly alter the key messages of this work as the principle we investigate is agnostic to topology used and our focus is on tree wide shifts in climate space occupation when Hutchinson's duality is considered. Rerunning any of our analyses also incurs a major issue of practicality. Due to changes in the institutional affiliations of the authors, we no longer have access to the computational resources required for the multi-month analysis times for such an endeavour, nor would this feasibly fit within the timeframe required by the editors. Without recourse for re-analysis, this is again why we do not heavily stress the biological reality or otherwise of the results based on the topology we opted for and keep the focus on the implications of the methods for inference of niche space from ancestral biogeography.

The phylogeny presented includes a number of taxa that are generally removed from the trees a priori because of their problematic nature of being highly incomplete (*Saltopus* is known from a mould) or potentially chimaeric (*Agnosphytis*), and so I would recommend a large portion of the tree be removed for this work. Ezcurra & Sues (2022) represents an updated iteration of this tree, and in their phylogenetic methods they list 40 taxa that were pruned from the analysis a priori as they are only used for disparity analyses. Many of these taxa are on the phylogeny figured, and so the phylogeny does bear a significant portion of taxa whose actual phylogenetic position is extremely challenging to place due to their incomplete nature. Additionally, Griffin et al., (2022) includes personal observations from Sterling Nesbitt (who described *Nyasasaurus*) whom attests that the materials of both *Agnosphytis* and *Nyasasaurus* are unlikely to reflect only two unique species.

Some of our taxa are poorly known, which will naturally introduce a greater degree of phylogenetic uncertainty into our trees. Phylogenetic placements will always be hypothetical, but the geographic presence of these fossils remains incontrovertible. As such, we believe that taxonomic inclusivity is the greater priority as the existence of even fragmentary taxa still provides essential data on the distribution of climates that early archosauromorphs occupied, which in turn informs their ancestral climate occupancy. This contrasts with a disparity analysis where anatomical completeness and so the availability of morphological data for a taxon is paramount and does give justification for exclusion of fragmentary taxa by Ezcurra and Sues (2022).

Regarding the specific examples noted by the reviewer, we believe that it would only be suitable to make such a revision following a formal systematic review of the materials of *Nyasasaurus* and *Agnosphitys*. This may result in new terminals being added to the tree, but these would then be geographically and temporally redundant relative to the existing information contributed by our current OTUs, and so we again do not suspect that such a revision would have a major impact on our analyses.

Conceivably, removal of taxa or revision of their placement in future analysis may have a small effect on the precise patterns of climatic disparity recovered within individual clades, but these are unlikely to be drastic given that the higher relationships of early archosauromorph clades are relatively robust. As our focus is on tree-wide patterns, rather than the specifics of subclade niche dynamics, we assert that our niche-focused results and their implications will also remain robust to our inclusion of more phylogenetically uncertain taxa.

More generally, we posit that questions regarding the phylogenetic and taxonomic scope of our analyses will be of second order to the much greater spatiotemporal biases imposed by the fossil record itself. All these fragmentary taxa still come from the same geographic regions as better described taxa and so largely contribute the same information regarding the geographic distribution of niches through time. The fragmentary taxa may introduce some noise into our analyses, but we believe that these uncertainties should be embraced to ensure that the broader, tree-wide pattern remains data-rich.

Personally, I do not agree with using a time-scaling approach on a tree derived from equal-weights parsimony. Parsimony lacks inherently informative branch lengths, and it has been shown that a Bayesian tip-dating analysis produces results more consistent with stratigraphic reality than time-scaling (King 2021). I suggest that you build your supertree based on the results of tip-dated phylogenies from Bayesian inference. I have included references to tip-dated phylogenies published for early archosaurs and non-dinosaurian dinosauriformes (Muller et al 2023), dinosaurs (Griffin et al 2022), and pseudosuchians (Turner et al. 2017). Generating a tip-dated supertree from these results (using Muller et al 2023 as the core of the tree), would provide more informative branch lengths and may impact the duration of the estimated dispersal events.

While we agree that recent work has demonstrated that tip-dated Bayesian methods can outperform maximum parsimony, we disagree that re-analysis with this alternative methodology is necessary for several reasons.

Firstly, the taxa added to the formal phylogenetic scaffold would still fall in the same immediate topological positions when constructing the informal supertree, resulting in no substantial change for a moderate portion of the overall topology. Secondly, we take

conceptual issue with post-hoc timescaling of a tree that had already incorporated stratigraphic information from fossils into its topology. A formal Bayesian tip-dated tree could be directly used instead, but then this would mean excluding many fragmentary taxa which can only be included informally and time-scaled post hoc. As discussed above, we believe that inclusion of more fragmentary taxa through informal placement is also of substantial benefit to our analyses, favouring the use of a stratigraphy-free maximum parsimony tree with post-hoc timescaling to enable their inclusion.

Our third reason relates to the differences (or otherwise) in topology between an MP and a BI tree. We find, for the formal portions of our tree, that there is very little difference in topology compared to later analyses using both parsimony and Bayesian analyses (e.g., Ezcurra et al., 2020; Muller et al., 2023). This is also supported by previous authors directly comparing early archosauromorph trees using both methods (Turner et al., 2017), suggesting that the availability of morphological data and the spatiotemporal sampling of early archosauromorph diversity exerts much greater control over the resulting topology of a tree rather than the reconstruction method used. While Bayesian tip-dating may be more methodologically robust in principle, we believe that the minor practical differences in topology and branch lengths would not noticeably affect the tree-wide patterns of coupled geographic and climatic occupancy that are the focus of this paper.

Our fourth reason relates to the accuracy of divergence times between an MP and a BI tree for early archosauromorphs. Again, we argue that the fundamental sampling limitations of their fossil record result in trees with very comparable sets of divergence times between both methods. While the higher degree of stratigraphic congruence offered by Bayesian trees may be taken in one regard as evidence of their greater accuracy, it also essentially assumes that stratigraphic ‘gappiness’ is also uniformly distributed across the tree which in practise is unlikely to be the case, particularly for a fossil record as sporadic as those of archosauromorphs.

To ascertain the potential impacts of different time-scaling approaches, we have calculated two measures of fit, the stratigraphic consistency index (SCI) and the gap excess ratio (GER), for our favoured topology, and for a recently published Bayesian tip-dated tree which also examines early archosauromorph biogeography (Heath et al., 2025). We find that the fits of both trees are highly similar ($SCI_{\text{ours}} = 70\%$, $SCI_{\text{heath}} = 72\%$, $GER_{\text{ours}} = 79\%$, $GER_{\text{heath}} = 80\%$), supporting our assertion that deviations in divergence terms under an alternative topology would be of little consequence for our downstream results. Therefore, a formal re-analysis, aside from incurring a hefty and potentially insurmountable computational burden, is unlikely to meaningfully improve the manuscript as it currently stands. This supporting test has been briefly referenced in the Materials and Methods (L437-441).

2. Framing the traditional tree as having Lagerpetids and Silesaurids as monophyletic stem dinosaur clades is in conflict with the current accepted hypotheses surrounding both of these groups. The 'control' topology for which everything is tested against should have Silesaurids as a paraphyletic grade just outside of Dinosauria, and also should have Lagerpetids as an early-diverging pterosauiromorphs that are sister to Pterosauria, as is presented in Muller et al (2023). I think it is fine to keep the comparison where both clades are considered stem dinosaurs, but framing that as the contemporary hypothesis is a bit outdated.

We agree that lagerpetids are well-established as the sister group to pterosaurs and the main results we present in the manuscript are from a tree which conforms to this contemporary understanding. We already state that the tree with this orthodox position is our 'favoured' phylogenetic hypothesis (L108) and have reinforced this stance in the results (L100-102). Their now heterodox position in our source tree reflects the information available when that informal phylogeny was first constructed, hence why we produced contemporary topologies for our core results.

We appreciate, however, that our presentation of this source tree as the 'traditional' hypothesis potentially a little misleading, as it suggests that we view their outdated placement as stem dinosaurs as the orthodox viewpoint. We have revised our phrasing in the manuscript, as well as our description of the tree alterations, so that all references to the 'traditional' tree are now referred to as 'historic' to make it clear that we accept the contemporary position of lagerpetids as pterosauiromorphs (L99, 352, 399, 405, 1180).

Moreover, we assert that there is merit in retaining their position as a monophyletic group of stem dinosaurs in the source tree. This historic and now heterodox placement is still useful to show how their revision substantially reduces the divergence time and location uncertainty for pterosauiromorphs, recovering a South American radiation for avemetatarsalian clades more generally. In the same way that stratigraphic fit is used to assess a phylogenetic topology, geographic consistency conceivably provides an analogous method of assessing support, and so our historic and contemporary results together provide valuable comparative evidence to support their reassignment as pterosauiromorphs.

Regarding silesaurids, we argue that the topology of the group, along with their position within broader archosaurian phylogeny remains an open debate. Practically, the issue of monophyly versus paraphyly is again likely a minor issue in this case. The order of tips between our alternative topologies is identical, so the ordering of geographic information in relation to divergence times will also be the same. This results in virtually identical divergence times and locations within Silesauridae and amongst dinosauria between all four of our alternative hypotheses (Fig. S31-34), suggesting that trees with further alternative topologies would be highly unlikely to

recover results that diverge from our findings, or which would cause us to revise the conclusions we draw.

We also view the mixture of monophyletic and paraphyletic topologies between our alternative trees as providing more robust coverage of the silesaurid debate as it stands – there are Triassic archosaur workers who continue to regard them as a monophyletic clade that, subject to current knowledge, is most appropriately placed outside of dinosaurs until new anatomical evidence and phylogenetic analysis can strongly prove otherwise. As we would be hesitant to ascribe monophyletic over paraphyletic interrelationships (or vice versa) of silesaurids, we have again revised the language in the manuscript to clarify to the reader that we do not regard the source tree as a contemporary hypothesis or necessarily even as a ‘control’, but as a ‘historic’ alternative to the other trees we analyse. This way, as before, our chosen set of main results is based on the phylogeny that provides the best resolved divergence time and locations.

Minor comments:

Line 41: change to "if fossil data are"

Change made (L41)

Line 60: I recommend changing instances of "basal" to "early-diverging". The term basal implies some degree of primitiveness, but many of these earliest diverging taxa lived within five million years of one another, which would suggest that they had evolved for roughly the same amount of time. I understand if you do not wish to make this change, I just wanted to make a case for what I consider to be a more accurate term, and so I will not comment on any other instances of this.

This same issue was picked up on by reviewer 1 and we agree that we do not want to impose a sense of false narrative on our results. Change made throughout the text (L55, 62, 92, 101, 110, 123, 138, 353, 399, 407, 1178, 1187)

Line 67: change "while" to "although"

Change made (L63).

Line 114: Is this meant to be silesaurids and ornithischians or silesaurids and lagerpetids?

Change made to ‘lagerpetids’ from ‘ornithischians’ (L107)

Line 146: There is evidence that the Doswelliidae is a northern radiation sister to Proterochampsia. Bayesian analysis by Wynd et al (2019) and implied weights

parsimony analyses done by Ezcurra and Sues (2022) both recover the Doswelliidae as being sister to the genus Proterochampsa, the earliest diverging of the proterochampsid genera. Because implied weights and bayesian inference with gamma distributed rate categories are adjusting the impacts that characters have on the overall topology, they are both essentially treating characters as if they evolve at different rates. Because it is highly unlikely that the hundreds of traits in the phylogenetic analyses would all share the exact same rate and phyletic history, I argue that the models with rate heterogeneity are likely more accurate. However, Muller et al (2023) do not recover the same relationship in their analysis, though that may be in part because they were focusing more of their attention elsewhere on the tree. It may be interesting to include this as another heterodox topology to test your model against, simply because I would imagine that embedding a Laurasian radiation near the base of an otherwise gondwanan clade may require a more intermediate ancestral location and thus unique dispersal patterns. I only suggest this because the Eucrocopoda are currently represented as the sister taxon to Archosauria and so a shift in the ancestral location for that clade may also impact reconstructions for the earliest diverging archosaur species.

We agree that more extensive tests of alternative topologies could potentially lead to divergent biogeographic results for specific subclades and nodes. However, this singular change is again unlikely to have any substantial impact on the broader narrative that is the focus of this work. Combined with the hurdles we face in re-running any of our analyses, we believe that this problem is best left for future authors to address. Nonetheless, we have made reference to the potential for alternative further topologies in our Materials and Methods (L402-404).

Lines 255-258: I disagree with this statement given the fact that you cannot perform a molecular analysis on these samples, and so you are bound to what you have access to. Additionally, the reference given for Oyston simply frames the relationship as morphological vs molecular whereas the reality of paleontology is a question of likelihood (trending bayesian) and parsimony. King (2021) suggests that bayesian tip-dated trees are more stratigraphically congruent and thus may be a better framework for your kind of question.

We agree that the point is moot without access to molecular data. We have removed the statement as such.

Line 301: The acronym EPME is only used twice throughout the manuscript. I would writing the entire phrase if only to prevent your reader from needing to go back to remember what it stood for in the introduction.

We have opted to use the entire phrase to streamline the reading of the overall manuscript (L57, 194, 296, 1235).

Line 384: I might suggest citing someone more contemporary than Seeley. Perhaps an early career scientist where citations could have a beneficial impact on a budding career.

We have opted to retain the reference to Seeley as this is the seminal citation for our chosen topology, but have also added a more recent ECR reference which discusses the broader issue of higher level early dinosaur interrelationships (L401).

Fig 1) The first fossil appearance for Allokotosauria is deep in the Norian, I'm assuming affiliated with Trilophosauridae? This should be back in the anisian based on Shringasaurus (Sengupta et al 2017).

This was a figure construction error on our part – we had the fossil appearance age set for the wrong allokotosaurian tip and have corrected this to use the age for Shringasaurus.

Labels for which time periods the paleomaps are based off of would be helpful to better gauge exactly when these plots are in reference to.

We have added indications of the palaeomap ages to the figure caption for clarity. However, we thought it best to not add the map age directly to the plot to avoid confusion over node divergence ages, given that each map is a compromise chosen to represent node divergence locations distributed across several stages, rather than the one specific to the figured map (L1166).

"Crown Ornithischia" should just be "Ornithischia." A crown group refers to the least inclusive clade that includes all presently living members of the group, and so Ornithischia cannot be considered a crown group.

Change made

References

Ezcurra, Martín D., and Hans-Dieter Sues. "A re-assessment of the osteology and phylogenetic relationships of the enigmatic, large-headed reptile *Sphodrosaurus pennsylvanicus* (Late Triassic, Pennsylvania, USA) indicates archosauriform affinities." *Journal of Systematic Palaeontology* 19, no. 24 (2021): 1643-1677.

Ezcurra, Martín D., Lucas E. Fiorelli, Agustín G. Martinelli, Sebastián Rocher, M. Belén von Baczko, Miguel Ezpeleta, Jeremías RA Taborda, E. Martín Hechenleitner, M. Jimena

Trotteyn, and Julia B. Desojo. "Deep faunistic turnovers preceded the rise of dinosaurs in southwestern Pangaea." *Nature Ecology & Evolution* 1, no. 10 (2017): 1477-1483.

Griffin, Christopher T., Brenen M. Wynd, Darlington Munyikwa, Tim J. Broderick, Michel Zondo, Stephen Tolan, Max C. Langer, Sterling J. Nesbitt, and Hazel R. Taruvinga. "Africa's oldest dinosaurs reveal early suppression of dinosaur distribution." *Nature* 609, no. 7926 (2022): 313-319.

King, Benedict. "Bayesian tip-dated phylogenetics in paleontology: topological effects and stratigraphic fit." *Systematic Biology* 70, no. 2 (2021): 283-294.

Müller, Rodrigo T., Martín D. Ezcurra, Mauricio S. Garcia, Federico L. Agnolín, Michelle R. Stocker, Fernando E. Novas, Marina B. Soares, Alexander WA Kellner, and Sterling J. Nesbitt. "New reptile shows dinosaurs and pterosaurs evolved among diverse precursors." *Nature* 620, no. 7974 (2023): 589-594.

Sengupta, Saradee, Martín D. Ezcurra, and Saswati Bandyopadhyay. "A new horned and long-necked herbivorous stem-archosaur from the Middle Triassic of India." *Scientific reports* 7, no. 1 (2017): 8366.

Turner, Alan H., Adam C. Pritchard, and Nicholas J. Matzke. "Empirical and Bayesian approaches to fossil-only divergence times: a study across three reptile clades." *PLoS one* 12, no. 2 (2017): e0169885.

Wynd, Brenen M., Sterling J. Nesbitt, Michelle R. Stocker, and Andrew B. Heckert. "A detailed description of *Rugarhynchos sixmilensis*, gen. et comb. nov. (Archosauriformes, Proterochampsia), and cranial convergence in snout elongation across stem and crown archosaurs." *Journal of Vertebrate Paleontology* 39, no. 6 (2019): e1748042.

Response references

Heath, J., Cooper, N., Upchurch, P., Mannion, P. (2025). Accounting for sampling heterogeneity suggests a low paleolatitude origin for dinosaurs. *Current Biology*, doi: 10.1016/j.cub.2024.12.053

Turner, A., Pritchard, A., Matzke, N. (2017). Empirical and Bayesian approaches to fossil-only divergence times: A study across three reptile clades. *PLoS One*, 12, e0169885

Response to Reviewers Round 1

Reviewer comments are in black. Our responses are in red. All changes in response to reviewer comments are referenced by line number in this document and highlighted in the revised manuscript file in yellow

Reviewer #1 (Remarks to the Author):

Overview

This is an interesting paper that is important not only for its contribution to our understanding of the early evolution of archosauromorphs, but also because it presents a new analytical method for looking at biogeography, fossil record sampling, and climatic niche occupancy in deep time. I found the paper easy to follow (though see my specific comments below regarding some aspects of clarity and terminology). I would particularly like to complement the authors on the excellence of the discussion section which is very explicit and clear about the limitations and caveats that apply to the study, especially with regard to fossil record sampling. One could certainly argue about the validity of certain assumptions underpinning this work, and no doubt some other reviewers may choose to do so. However, any modelling approach of this type must inevitably balance realism against pragmatism. Moreover, this clearly represents an advanced in an interesting direction and is something that I believe other researchers will want to see and use. I am therefore happy to support publication of this work subject to the authors addressing the issues I have set out below.

We thank the reviewer for their positive outlook on the paper and have made every effort to incorporate their feedback into the revised manuscript.

Specific points

One of the conclusions from this paper is to support the long-held view that dinosaurs originated in South America. However, as the authors note themselves, there are concerns regarding incomplete fossil record sampling, and this may well have affected our views on dinosaur origins. In particular, a recent paper which came out while the current study was in review (Lovelace et al 2025 Rethinking dinosaur origins... Zoological Journal of the Linnean society), reports an interesting low palaeolatitude fauna from the late Carnian of North America. I am not asking the authors to re-run any of their analyses, but I think they should acknowledge the existence of this paper and compare its implications with their inferences – this might help illustrate some of the limitations and caveats they note in their discussion section.

We are grateful to the reviewer for drawing our attention to this paper. We have incorporated it into our discussion (L252-256), along with another paper out around the same time that explicitly models a lower latitude origin for dinosaurs, and believe that the alternative results they recover strengthen our message that biogeographic origins are heavily contingent on sampling in the fossil record.

Lines 66-67 – “(Dunne et al., 2021, 2023). These narratives, however, are based on inferences from their sporadically sampled fossil record and basal phylogenetic relationships. Further, while early...”.

I recommend changing “basal” to “early diverging” because of the problems associated with “basal” in a phylogenetic context – see Omland et al. (2008)

Omland, K. E., L. G. Cook, and M. D. Crisp. 2008. Tree thinking for all biology: the problem with reading phylogenies as ladders of progress. *BioEssays* 30:854–867.

Change made to ‘early diverging’ here and throughout the rest of the manuscript (L55, 62, 92, 101, 110, 123, 138, 353, 399, 407, 1178, 1187)

Also, “Narratives” is potentially problematic and I would recommend changing this to something like “scenarios” or even “hypotheses”. “Narratives” in the context of biogeography has come to mean an account based largely on a literal reading of the fossil record (see Ball 1975 and papers that have cited this subsequently). However, the various studies cited at this point in the paper include many that have made decent attempts to take sampling into account and fill in gaps using a variety of techniques including those based on a phylogenetic framework. The authors may well think that these techniques are flawed in some way or could be improved upon, but I think it will muddy the waters somewhat if they describe the results as “narratives”.

Ball, I. R. 1975. Nature and formulation of biogeographic hypotheses. *Systematic Zoology*, 24, 407–430.

Change made to ‘scenarios’ (L61)

Lines 106-114 - “...in western Gondwana). Originations within the archosauromorph crown group are sensitive to the placement of traditional stem dinosaur clades (Fig. 2). Placement of lagerpetids as the sister group of pterosaurs (Ezcurra et al., 2020), and silesaurids as a basal grade of ornithischian dinosaurs (Muller and Garcia 2020) collapses poorly resolved bimodal estimates for the geographic origins of archosaurs and various pseudosuchian clades to more robust unimodal estimates (Fig. 2A, B, Supplementary Data 1), and substantially reduces divergence time uncertainty for pterosauromorphs, pterosaurs and ornithischians (Fig. 2C, D). All subsequent results presented here are based on a topology with revised positions of both silesaurids and

ornithischians”.

Again, avoid “basal” in the context of phylogenetic relationships.

Addressed as above

Also, I think the last line should read “All subsequent results presented here are based on a topology with revised positions of both silesaurids and lagerpetids”. As you can see, the previous sentences in lines 106-114 talk about two aspects of topology – the positions of silesaurids and lagerpetids, but do not talk about the position of ornithischians. So the final line as originally written is a bit confusing because there has been no previous reference to the position of Ornithischia and I can only make sense of it if I substitute “lagerpetids” for “ornithischians”.

This was a typo on our part and the change has been made from ‘ornithischians’ to ‘lagerpetids’ (L107)

Lines 474-476 – “...of highest density cell in the grid, and the 95% highest density intervals for the estimated longitudes and latitudes.”

Here, and at several other points, I think that “latitude” and “longitude” should actually be “palaeolatitude” and “Palaeolongitude”, respectively. Each fossil specimen has a modern day latitude and longitude where it was found, and an almost certainly different Palaeolatitude and palaeolongitude where the living animal existed. If we are talking about estimating the location of an ancestor, for example, then this should be expressed as palaeolatitude/Palaeolongitude in order to keep this distinct from any confusion with the location of fossils today. Not only is this important when describing evolutionary scenarios in deep time, but also in any study that touches on sampling biases. Sampling bias may relate to events that happened at a particular place in the past (e.g. geological factors such as sedimentation and erosion - in which case discussing their palaeo location is relevant), or they may relate to modern day biases (e.g. those caused by colonialism, other anthropogenic biases, difficulty in working in certain harsh environments and So on – in which case the terms latitude and longitude would help keep it clear that we were talking about the present time). So, I think the authors need to go through the manuscript carefully and ensure that they use the appropriate terminology depending on the context.

We thank the reviewer for highlighting this potential source of confusion for readers and agree with their point. We have gone through the manuscript and implemented these changes throughout (L253, 448, 498, 504, 507, 515, 554, 646)

Paul Upchurch

Reviewer #2 (Remarks to the Author):

This is an ambitious paper that sets out to track the spatial origins and biogeographic movements of the earliest archosauromorphs through the use of phylogenetic niche modeling and a newly developed spatiotemporal phylogeographic path analysis method. The paper was somewhat difficult for me as my scientific expertise is in phylogeography and niche modeling, while my knowledge of paleontology is more that of an interested layman (for instance, I have the major geological time periods memorized, but not the smaller stages within them). Nonetheless, I was overall quite impressed by the paper and its methodology. My critique is fairly limited. Mostly I wish the authors to better integrate certain topics into the discussion, and to simplify the wording of certain ideas to make them not just easier to understand but make it easier to see how they affect the big picture and the results overall. For instance, in Figure 6, it is not entirely clearly from the text what the biological and ecological meanings of the displayed statistics are, and how the differences between the ways they were calculated (i.e., the three different colored lines in each plot) matter (again, in a bio/ecological context). This is all probably clear to the author(s), but in my first pass through (which is the very most you can hope for from most readers), these nuances didn't really work their way into my brain. I would also like to see a deeper discussion of caveats and issues related to the phylogenetic niche modeling part of the paper. Overall, I commend the authors on their ambitious undertaking and appreciate being sent this paper that combines a professional interest (niche modeling) with a casual, more fanboy-ish one (dinosaurs!).

We thank the reviewer for their positive response to the manuscript. Following their feedback we have carefully checked our phrasing throughout the manuscript to ensure that it is accessible as possible for readers across different disciplines.

Major comments:

The discussion of caveats relating to potential sampling bias and the geo model was well done, but I was hoping for more discussion of the caveats of the niche modeling method itself, both in terms of the climate and occurrence data used, and the modeling methodology. Many readers such as myself that come from a niche modeler/phylogeography background vs. a paleontological one will desire a deeper discussion of this topic.

We have added a new paragraph to our discussion to give a more thorough treatment of the limitations of our climate simulation and occurrence data with regards to niche modelling and dispersal pathway estimation (L325-348)

Lines 539-541: Here it is stated that nodal climate conditions were estimated with ML using "extracted climatic conditions for the fossil tips." What exactly does this mean? Specifically, I would be interested to know what the actual values being used in the ancestral state reconstruction are. I think this should be stated more clearly for the reader overall.

We have rephrased the text to make the source of the climatic tip states absolutely clear (L562-563).

I wish the differences between climate space occupancy derived from tips & ML estimates, tips & ancestral locations, and ancestral locations & dispersal paths was explained better prior to or early on in the Results, both in terms of how these values are calculated and what they mean biologically compared to each other.

We have revised our presentation of the climatic disparity results so that the biological meanings of our summary statistics are clearly defined prior to describing any trends (L178-187). Specifically, we now refer to the diversity and consistency of occupied climatic niches based on the range and density of dispersal pathways in climate space. We have also reworked the caption for Fig. 6 so that these meanings are also conveyed to the figure, rather than requiring the reader to flick between text and figure (L1225-1234)

Minor comments:

Lines 174-185: I recommend continuing to provide references to Fig 5 throughout this paragraph (as with line 177) to make it easier for the reader to follow along.

Additional references to Fig. 5 have been added throughout the paragraph (L171, 174, 177)

Line 247: I'm not sure what "once geographically proximate" means here.

We have changed 'proximate' to 'closer' to keep our meaning clear (L241)

Lines 880-881: I believe this should be referring to Figure 3.

A typo on our part. Corrected to Fig. 3 (L166)

Line 919: I believe this should refer to Supplementary Data 3.

This was another typo. We have elected to convert this particular file to an Extended Data Figure and have updated the text accordingly (L141, 1206)

I recommend making font sizes larger in many figures, such as Fig. 3. Many of these labels are too small to read at normal sizes.

We agree that our labels were quite small in many cases. Where possible we have enlarged the text to address these issues, although in some cases this was not always possible (notably the tip labels for the phylogeny). With regards to this specific instance, this is also why we thought it prudent to present a simplified version of the phylogeny in Fig. 1. This way a reader can still get a clear sense of the taxonomic scope of the tree, then refer to Fig. 3 in the online version for the specific taxa included.

Reviewer #3 (Remarks to the Author):

The authors present a novel approach to reconstructing the biogeographic history of long extinct lineages. By utilizing the geographic and temporal locations of species within a greater clade, the authors estimate clade-wide dispersal maps for each stage from the end of the Early Permian (Kungurian) to the end of the Triassic (Rhaetian), based on recent phylogenies. This work is interesting and provides a unique perspective on reconstructing ancient biogeographic hypotheses at an unprecedented level of detail. This work should certainly be published, and I believe that *Nature Ecology & Evolution* would be a strong fit for this work. With that being said, I believe that the underlying phylogeny may not be the best to address this question. The topology presented is based off of an analysis from 2017 that has had numerous modifications to it in the intervening years, some of which include topologies that are contradictory to the narrative presented herein. Most notable of which are the Lagerpetid avemetatarsalians which are represented as having an orthodox placement as a monophyletic clade among the dinosaurian stem. The authors represent a heterodox relationship for Lagerpetids as the sister taxon to Pterosauria. However, there have been a number of papers within the last few years that have presented Lagerpetids as the sister taxon to Pterosauria, with increasing confidence in each publication. I think it would benefit the others to reframe their argument to match the currently accepted topologies for these groups, some of which have been published quite recently. Additionally, there are existing arguments that suggest that bayesian tip-dated phylogenies are more appropriate than time-scaled parsimonious trees, as the bayesian estimations tend to more closely match the given stratigraphic records. Given that there have been a series of recent publications that have performed both parsimony and tip-dated bayesian inference on archosauromorphs, dinosaurs, and crocodiles, I think it would likely benefit the authors to incorporate a tip-dated bayesian supertree with the most contemporary topologies either alongside or in lieu of the

existing topology. I understand this is a big ask, but I think doing so has the potential to provide even more clarity and resolution to this work.

We are grateful to the reviewer for their feedback on our manuscript and appreciate the concerns raised by advances made in time-scaling methodologies and early archosauromorph topologies. The reviewer's points have given us a valuable opportunity to reflect on the limitations of our work and as such we have given them greater emphasis in our discussion to ensure that the phylogenetic caveats are abundantly clear (L252-256), particularly the potential for divergent biogeographic results as demonstrated by very recently published work on early dinosaurian origins (Heath et al., 2025).

The critical issue is whether a revised tree would make a significant practical difference to the results and conclusions we currently present. While we agree in principle that Bayesian tip-dated methods may be preferable to parsimony with post-hoc time-scaling, we do not think that the substantial task of re-running out analyses with a revised tree would incur significant gain in terms of either the overall accuracy of our findings or the implications we draw from them. This partly stems from our assertion that the differences of taxon sampling, topology and branch lengths will be small enough so as not to alter message of our paper.

Ultimately, our goal is to present a novel approach to tackle the issue of incompletely sampled niches in the fossil record, using tree wide results rather than the dynamics of specific subclades. Where we do discuss the biogeographic origins of archosauromorphs, this is mainly to provide a scaffold for our subsequent investigation of Hutchinson's duality across the entire tree. In turn this is why we continue to carefully highlight the sensitivity of any biogeographic analysis to topological choice and new fossil discoveries, rather than presenting our specific findings as an incontrovertible reconstruction of their past biogeography.

Given the undeniable importance of methodological and topological debate the reviewer points out, we have provided an extensive and thorough reasoning below to justify our adherence to our current topology and methods. In addition, we would also face major difficulties in rerunning any analyses due to unavailability of the required computational resources, as a secondary reason for retaining the results presented herein.

If you have any questions regarding my review, I would be happy to chat with you more in depth.

Best,
Brenen Wynd

Remarks to the author:

Major comments:

1. I believe that there are some core issues to the phylogenetic materials chosen to base the supertree off of that need to be addressed.

Ultimately, we do not believe that re-analysis using an alternative supertree would strongly alter the key messages of this work as the principle we investigate is agnostic to topology used and our focus is on tree wide shifts in climate space occupation when Hutchinson's duality is considered. Rerunning any of our analyses also incurs a major issue of practicality. Due to changes in the institutional affiliations of the authors, we no longer have access to the computational resources required for the multi-month analysis times for such an endeavour, nor would this feasibly fit within the timeframe required by the editors. Without recourse for re-analysis, this is again why we do not heavily stress the biological reality or otherwise of the results based on the topology we opted for and keep the focus on the implications of the methods for inference of niche space from ancestral biogeography.

The phylogeny presented includes a number of taxa that are generally removed from the trees a priori because of their problematic nature of being highly incomplete (*Saltopus* is known from a mould) or potentially chimaeric (*Agnosphytis*), and so I would recommend a large portion of the tree be removed for this work. Ezcurra & Sues (2022) represents an updated iteration of this tree, and in their phylogenetic methods they list 40 taxa that were pruned from the analysis a priori as they are only used for disparity analyses. Many of these taxa are on the phylogeny figured, and so the phylogeny does bear a significant portion of taxa whose actual phylogenetic position is extremely challenging to place due to their incomplete nature. Additionally, Griffin et al., (2022) includes personal observations from Sterling Nesbitt (who described *Nyasasaurus*) whom attests that the materials of both *Agnosphytis* and *Nyasasaurus* are unlikely to reflect only two unique species.

Some of our taxa are poorly known, which will naturally introduce a greater degree of phylogenetic uncertainty into our trees. Phylogenetic placements will always be hypothetical, but the geographic presence of these fossils remains incontrovertible. As such, we believe that taxonomic inclusivity is the greater priority as the existence of even fragmentary taxa still provides essential data on the distribution of climates that early archosauromorphs occupied, which in turn informs their ancestral climate occupancy. This contrasts with a disparity analysis where anatomical completeness and so the availability of morphological data for a taxon is paramount and does give justification for exclusion of fragmentary taxa by Ezcurra and Sues (2022).

Regarding the specific examples noted by the reviewer, we believe that it would only be suitable to make such a revision following a formal systematic review of the materials of *Nyasasaurus* and *Agnosphitys*. This may result in new terminals being added to the tree, but these would then be geographically and temporally redundant relative to the existing information contributed by our current OTUs, and so we again do not suspect that such a revision would have a major impact on our analyses.

Conceivably, removal of taxa or revision of their placement in future analysis may have a small effect on the precise patterns of climatic disparity recovered within individual clades, but these are unlikely to be drastic given that the higher relationships of early archosauromorph clades are relatively robust. As our focus is on tree-wide patterns, rather than the specifics of subclade niche dynamics, we assert that our niche-focused results and their implications will also remain robust to our inclusion of more phylogenetically uncertain taxa.

More generally, we posit that questions regarding the phylogenetic and taxonomic scope of our analyses will be of second order to the much greater spatiotemporal biases imposed by the fossil record itself. All these fragmentary taxa still come from the same geographic regions as better described taxa and so largely contribute the same information regarding the geographic distribution of niches through time. The fragmentary taxa may introduce some noise into our analyses, but we believe that these uncertainties should be embraced to ensure that the broader, tree-wide pattern remains data-rich.

Personally, I do not agree with using a time-scaling approach on a tree derived from equal-weights parsimony. Parsimony lacks inherently informative branch lengths, and it has been shown that a Bayesian tip-dating analysis produces results more consistent with stratigraphic reality than time-scaling (King 2021). I suggest that you build your supertree based on the results of tip-dated phylogenies from Bayesian inference. I have included references to tip-dated phylogenies published for early archosaurs and non-dinosaurian dinosauriformes (Muller et al 2023), dinosaurs (Griffin et al 2022), and pseudosuchians (Turner et al. 2017). Generating a tip-dated supertree from these results (using Muller et al 2023 as the core of the tree), would provide more informative branch lengths and may impact the duration of the estimated dispersal events.

While we agree that recent work has demonstrated that tip-dated Bayesian methods can outperform maximum parsimony, we disagree that re-analysis with this alternative methodology is necessary for several reasons.

Firstly, the taxa added to the formal phylogenetic scaffold would still fall in the same immediate topological positions when constructing the informal supertree, resulting in no substantial change for a moderate portion of the overall topology. Secondly, we take

conceptual issue with post-hoc timescaling of a tree that had already incorporated stratigraphic information from fossils into its topology. A formal Bayesian tip-dated tree could be directly used instead, but then this would mean excluding many fragmentary taxa which can only be included informally and time-scaled post hoc. As discussed above, we believe that inclusion of more fragmentary taxa through informal placement is also of substantial benefit to our analyses, favouring the use of a stratigraphy-free maximum parsimony tree with post-hoc timescaling to enable their inclusion.

Our third reason relates to the differences (or otherwise) in topology between an MP and a BI tree. We find, for the formal portions of our tree, that there is very little difference in topology compared to later analyses using both parsimony and Bayesian analyses (e.g., Ezcurra et al., 2020; Muller et al., 2023). This is also supported by previous authors directly comparing early archosauromorph trees using both methods (Turner et al., 2017), suggesting that the availability of morphological data and the spatiotemporal sampling of early archosauromorph diversity exerts much greater control over the resulting topology of a tree rather than the reconstruction method used. While Bayesian tip-dating may be more methodologically robust in principle, we believe that the minor practical differences in topology and branch lengths would not noticeably affect the tree-wide patterns of coupled geographic and climatic occupancy that are the focus of this paper.

Our fourth reason relates to the accuracy of divergence times between an MP and a BI tree for early archosauromorphs. Again, we argue that the fundamental sampling limitations of their fossil record result in trees with very comparable sets of divergence times between both methods. While the higher degree of stratigraphic congruence offered by Bayesian trees may be taken in one regard as evidence of their greater accuracy, it also essentially assumes that stratigraphic ‘gappiness’ is also uniformly distributed across the tree which in practise is unlikely to be the case, particularly for a fossil record as sporadic as those of archosauromorphs.

To ascertain the potential impacts of different time-scaling approaches, we have calculated two measures of fit, the stratigraphic consistency index (SCI) and the gap excess ratio (GER), for our favoured topology, and for a recently published Bayesian tip-dated tree which also examines early archosauromorph biogeography (Heath et al., 2025). We find that the fits of both trees are highly similar ($SCI_{\text{ours}} = 70\%$, $SCI_{\text{heath}} = 72\%$, $GER_{\text{ours}} = 79\%$, $GER_{\text{heath}} = 80\%$), supporting our assertion that deviations in divergence terms under an alternative topology would be of little consequence for our downstream results. Therefore, a formal re-analysis, aside from incurring a hefty and potentially insurmountable computational burden, is unlikely to meaningfully improve the manuscript as it currently stands. This supporting test has been briefly referenced in the Materials and Methods (L437-441).

2. Framing the traditional tree as having Lagerpetids and Silesaurids as monophyletic stem dinosaur clades is in conflict with the current accepted hypotheses surrounding both of these groups. The 'control' topology for which everything is tested against should have Silesaurids as a paraphyletic grade just outside of Dinosauria, and also should have Lagerpetids as an early-diverging pterosauiromorphs that are sister to Pterosauria, as is presented in Muller et al (2023). I think it is fine to keep the comparison where both clades are considered stem dinosaurs, but framing that as the contemporary hypothesis is a bit outdated.

We agree that lagerpetids are well-established as the sister group to pterosaurs and the main results we present in the manuscript are from a tree which conforms to this contemporary understanding. We already state that the tree with this orthodox position is our 'favoured' phylogenetic hypothesis (L108) and have reinforced this stance in the results (L100-102). Their now heterodox position in our source tree reflects the information available when that informal phylogeny was first constructed, hence why we produced contemporary topologies for our core results.

We appreciate, however, that our presentation of this source tree as the 'traditional' hypothesis potentially a little misleading, as it suggests that we view their outdated placement as stem dinosaurs as the orthodox viewpoint. We have revised our phrasing in the manuscript, as well as our description of the tree alterations, so that all references to the 'traditional' tree are now referred to as 'historic' to make it clear that we accept the contemporary position of lagerpetids as pterosauiromorphs (L99, 352, 399, 405, 1180).

Moreover, we assert that there is merit in retaining their position as a monophyletic group of stem dinosaurs in the source tree. This historic and now heterodox placement is still useful to show how their revision substantially reduces the divergence time and location uncertainty for pterosauiromorphs, recovering a South American radiation for avemetatarsalian clades more generally. In the same way that stratigraphic fit is used to assess a phylogenetic topology, geographic consistency conceivably provides an analogous method of assessing support, and so our historic and contemporary results together provide valuable comparative evidence to support their reassignment as pterosauiromorphs.

Regarding silesaurids, we argue that the topology of the group, along with their position within broader archosaurian phylogeny remains an open debate. Practically, the issue of monophyly versus paraphyly is again likely a minor issue in this case. The order of tips between our alternative topologies is identical, so the ordering of geographic information in relation to divergence times will also be the same. This results in virtually identical divergence times and locations within Silesauridae and amongst dinosauria between all four of our alternative hypotheses (Fig. S31-34), suggesting that trees with further alternative topologies would be highly unlikely to

recover results that diverge from our findings, or which would cause us to revise the conclusions we draw.

We also view the mixture of monophyletic and paraphyletic topologies between our alternative trees as providing more robust coverage of the silesaurid debate as it stands – there are Triassic archosaur workers who continue to regard them as a monophyletic clade that, subject to current knowledge, is most appropriately placed outside of dinosaurs until new anatomical evidence and phylogenetic analysis can strongly prove otherwise. As we would be hesitant to ascribe monophyletic over paraphyletic interrelationships (or vice versa) of silesaurids, we have again revised the language in the manuscript to clarify to the reader that we do not regard the source tree as a contemporary hypothesis or necessarily even as a ‘control’, but as a ‘historic’ alternative to the other trees we analyse. This way, as before, our chosen set of main results is based on the phylogeny that provides the best resolved divergence time and locations.

Minor comments:

Line 41: change to "if fossil data are"

Change made (L41)

Line 60: I recommend changing instances of "basal" to "early-diverging". The term basal implies some degree of primitiveness, but many of these earliest diverging taxa lived within five million years of one another, which would suggest that they had evolved for roughly the same amount of time. I understand if you do not wish to make this change, I just wanted to make a case for what I consider to be a more accurate term, and so I will not comment on any other instances of this.

This same issue was picked up on by reviewer 1 and we agree that we do not want to impose a sense of false narrative on our results. Change made throughout the text (L55, 62, 92, 101, 110, 123, 138, 353, 399, 407, 1178, 1187)

Line 67: change "while" to "although"

Change made (L63).

Line 114: Is this meant to be silesaurids and ornithischians or silesaurids and lagerpetids?

Change made to ‘lagerpetids’ from ‘ornithischians’ (L107)

Line 146: There is evidence that the Doswelliidae is a northern radiation sister to Proterochampsia. Bayesian analysis by Wynd et al (2019) and implied weights

parsimony analyses done by Ezcurra and Sues (2022) both recover the Doswelliidae as being sister to the genus Proterochampsa, the earliest diverging of the proterochampsid genera. Because implied weights and bayesian inference with gamma distributed rate categories are adjusting the impacts that characters have on the overall topology, they are both essentially treating characters as if they evolve at different rates. Because it is highly unlikely that the hundreds of traits in the phylogenetic analyses would all share the exact same rate and phyletic history, I argue that the models with rate heterogeneity are likely more accurate. However, Muller et al (2023) do not recover the same relationship in their analysis, though that may be in part because they were focusing more of their attention elsewhere on the tree. It may be interesting to include this as another heterodox topology to test your model against, simply because I would imagine that embedding a Laurasian radiation near the base of an otherwise gondwanan clade may require a more intermediate ancestral location and thus unique dispersal patterns. I only suggest this because the Eucrocopoda are currently represented as the sister taxon to Archosauria and so a shift in the ancestral location for that clade may also impact reconstructions for the earliest diverging archosaur species.

We agree that more extensive tests of alternative topologies could potentially lead to divergent biogeographic results for specific subclades and nodes. However, this singular change is again unlikely to have any substantial impact on the broader narrative that is the focus of this work. Combined with the hurdles we face in re-running any of our analyses, we believe that this problem is best left for future authors to address. Nonetheless, we have made reference to the potential for alternative further topologies in our Materials and Methods (L402-404).

Lines 255-258: I disagree with this statement given the fact that you cannot perform a molecular analysis on these samples, and so you are bound to what you have access to. Additionally, the reference given for Oyston simply frames the relationship as morphological vs molecular whereas the reality of paleontology is a question of likelihood (trending bayesian) and parsimony. King (2021) suggests that bayesian tip-dated trees are more stratigraphically congruent and thus may be a better framework for your kind of question.

We agree that the point is moot without access to molecular data. We have removed the statement as such.

Line 301: The acronym EPME is only used twice throughout the manuscript. I would writing the entire phrase if only to prevent your reader from needing to go back to remember what it stood for in the introduction.

We have opted to use the entire phrase to streamline the reading of the overall manuscript (L57, 194, 296, 1235).

Line 384: I might suggest citing someone more contemporary than Seeley. Perhaps an early career scientist where citations could have a beneficial impact on a budding career.

We have opted to retain the reference to Seeley as this is the seminal citation for our chosen topology, but have also added a more recent ECR reference which discusses the broader issue of higher level early dinosaur interrelationships (L401).

Fig 1) The first fossil appearance for Allokotosauria is deep in the Norian, I'm assuming affiliated with Trilophosauridae? This should be back in the anisian based on Shringasaurus (Sengupta et al 2017).

This was a figure construction error on our part – we had the fossil appearance age set for the wrong allokotosaurian tip and have corrected this to use the age for Shringasaurus.

Labels for which time periods the paleomaps are based off of would be helpful to better gauge exactly when these plots are in reference to.

We have added indications of the palaeomap ages to the figure caption for clarity. However, we thought it best to not add the map age directly to the plot to avoid confusion over node divergence ages, given that each map is a compromise chosen to represent node divergence locations distributed across several stages, rather than the one specific to the figured map (L1166).

"Crown Ornithischia" should just be "Ornithischia." A crown group refers to the least inclusive clade that includes all presently living members of the group, and so Ornithischia cannot be considered a crown group.

Change made

References

Ezcurra, Martín D., and Hans-Dieter Sues. "A re-assessment of the osteology and phylogenetic relationships of the enigmatic, large-headed reptile *Sphodrosaurus pennsylvanicus* (Late Triassic, Pennsylvania, USA) indicates archosauriform affinities." *Journal of Systematic Palaeontology* 19, no. 24 (2021): 1643-1677.

Ezcurra, Martín D., Lucas E. Fiorelli, Agustín G. Martinelli, Sebastián Rocher, M. Belén von Baczko, Miguel Ezpeleta, Jeremías RA Taborda, E. Martín Hechenleitner, M. Jimena

Trotteyn, and Julia B. Desojo. "Deep faunistic turnovers preceded the rise of dinosaurs in southwestern Pangaea." *Nature Ecology & Evolution* 1, no. 10 (2017): 1477-1483.

Griffin, Christopher T., Brenen M. Wynd, Darlington Munyikwa, Tim J. Broderick, Michel Zondo, Stephen Tolan, Max C. Langer, Sterling J. Nesbitt, and Hazel R. Taruvinga. "Africa's oldest dinosaurs reveal early suppression of dinosaur distribution." *Nature* 609, no. 7926 (2022): 313-319.

King, Benedict. "Bayesian tip-dated phylogenetics in paleontology: topological effects and stratigraphic fit." *Systematic Biology* 70, no. 2 (2021): 283-294.

Müller, Rodrigo T., Martín D. Ezcurra, Mauricio S. Garcia, Federico L. Agnolín, Michelle R. Stocker, Fernando E. Novas, Marina B. Soares, Alexander WA Kellner, and Sterling J. Nesbitt. "New reptile shows dinosaurs and pterosaurs evolved among diverse precursors." *Nature* 620, no. 7974 (2023): 589-594.

Sengupta, Saradee, Martín D. Ezcurra, and Saswati Bandyopadhyay. "A new horned and long-necked herbivorous stem-archosaur from the Middle Triassic of India." *Scientific reports* 7, no. 1 (2017): 8366.

Turner, Alan H., Adam C. Pritchard, and Nicholas J. Matzke. "Empirical and Bayesian approaches to fossil-only divergence times: a study across three reptile clades." *PloS one* 12, no. 2 (2017): e0169885.

Wynd, Brenen M., Sterling J. Nesbitt, Michelle R. Stocker, and Andrew B. Heckert. "A detailed description of *Rugarhynchos sixmilensis*, gen. et comb. nov. (Archosauriformes, Proterochampsia), and cranial convergence in snout elongation across stem and crown archosaurs." *Journal of Vertebrate Paleontology* 39, no. 6 (2019): e1748042.

Response references

Heath, J., Cooper, N., Upchurch, P., Mannion, P. (2025). Accounting for sampling heterogeneity suggests a low paleolatitude origin for dinosaurs. *Current Biology*, doi: 10.1016/j.cub.2024.12.053

Turner, A., Pritchard, A., Matzke, N. (2017). Empirical and Bayesian approaches to fossil-only divergence times: A study across three reptile clades. *PLoS One*, 12, e0169885

Response to Reviewers Round 2

Reviewer comments are in black. Our responses are in red. All changes in response to reviewer comments are referenced by line number in this document and highlighted in the revised manuscript file in yellow

Reviewer #1 (Remarks to the Author):

Review of revised version

I have been through the revised manuscript and the rebuttal letter. I note that the authors have made all of the changes I requested. Reviewers 3 and 4 raise issues about the supertree used (Bayesian versus Parsimony), and advocate for the use of Bayesian trees which in turn would have necessitated rerunning all analyses. I completely agree with the authors that this is something they should not be compelled to do. Aside from the pragmatic issues of obtaining computing time noted by the authors, I agree with the suggestion that such a major change is unlikely to radically alter their conclusions. Moreover, for me, the importance of this paper lies more in the methodological novelty (which will have a strong impact on the field by providing a new set of techniques for the rest of us to play with), rather than the specifics of archosauromorph biogeography. There are certainly some newsworthy and interesting aspects to what the paper says about archosauromorphs, but fundamentally the results are not too surprising in that regard – it is the methodological novelty that makes this paper particularly important and why I believe it should be published. Therefore, based on the revised manuscript and the arguments made in the rebuttal letter, I would now support acceptance of the paper in its current form.

Thanks

Reviewer #2 (Remarks to the Author):

Thank you to the authors for addressing my few comments, which mostly addressed a few topics I wanted additional clarity on.

L325-348: I thought the new paragraph added to the discussion about the limitations of the niche modeling approach was well done and addressed my desire for proper discussion of this topic.

Thanks

L562-563: Thank you for clarifying what the source of the climatic tip states is.

Thanks

L178-187: The new explanation of the various ways you quantified archosauromorph niches is helpful, thank you.

Thanks

The new version of the manuscript is ready to go. I commend the authors on their work.

Thanks

Reviewer #2 (Remarks on code availability):

n/a

Thanks

Reviewer #3 (Remarks to the Author):

The authors took the critiques of myself and the other reviewers and offered well-reasoned responses in cases where comments weren't feasible to be incorporated. I think the authors responses and associated corrections made to the manuscript warrant publication in Nature Ecology & Evolution. The inclusion of SCI and GER tree statistics provides a useful justification for the choice of tree used herein. Additionally, the language surrounding phylogenetic and systematic relationships are more clear.

Thanks

Response to the response:

"Secondly, we take conceptual issue with post-hoc timescaling of a tree that had already incorporated stratigraphic information from fossils into its topology."

Apologies for my poor phrasing, I was not intending to recommend timescaling a bayesian tree. My intent was that the construction of a supertree from established bayesian topologies would remove the need for post-hoc scaling. I agree that time-scaling a bayesian topology would be both redundant and counter-productive.

Thanks

Minor comment:

Line 402: Change to "We also elected to retain the interrelationships of eucrocopodans and early-diverging pseudosuchians in our source tree,..."

Doswelliid arguments for being psuedosuchian are not supported today. Including Eucrocopoda better encapsulates relationships within and outside of Archosauria.

Change made

Reviewer #3 (Remarks on code availability):

I include comments regarding the code availability in confidential remarks to the editor. But as far as I can tell (and I tried searching the websites mentioned in the Data and Code availability sections), the data and code are not presently available or made public to anyone but the authors. If this is not the case, then there are not appropriate links or avenues to reasonably and easily find the supplementary information for this project.

We re-tested our links to ensure that they were accessible by other parties. Those links are supplied in the data and code availability section